# Global Optimization Algorithm through High-Resolution Sampling

**Daniel Cortild**                                                    *d.cortild@rug.nl*
*University of Groningen, Netherlands*
*Laboratoire de Finance des Marchés de l'Energie, Dauphine, CREST, EDF R&D, France*

**Claire Delplancke**                                               *claire.delplancke@edf.fr*
*EDF R&D Palaiseau, France*

**Nadia Oudjane**                                                   *nadia.oudjane@edf.fr*
*Laboratoire de Finance des Marchés de l'Energie, Dauphine, CREST, EDF R&D, France*

**Juan Peypouquet**                                                 *j.g.peypouquet@rug.nl*
*University of Groningen, Netherlands*

**Reviewed on OpenReview:** *https://openreview.net/forum?id=r3VEA1AWY5*

## Abstract

We present an optimization algorithm that can identify a global minimum of a potentially nonconvex smooth function with high probability, assuming the Gibbs measure of the potential satisfies a logarithmic Sobolev inequality. Our contribution is twofold: on the one hand we propose said global optimization method, which is built on an oracle sampling algorithm producing arbitrarily accurate samples from a given Gibbs measure. On the other hand, we propose a new sampling algorithm, drawing inspiration from both overdamped and underdamped Langevin dynamics, as well as from the high-resolution differential equation known for its acceleration in deterministic settings. While the focus of the paper is primarily theoretical, we demonstrate the effectiveness of our algorithms on the Rastrigin function, where it outperforms recent approaches.

## 1 Introduction

Smooth nonconvex optimization remains a challenge with broad applications in machine learning and statistical inference. Despite significant advances, convex optimization techniques often lead to suboptimal or computationally infeasible solutions in many inherently nonconvex real-world problems.

In this paper, we focus on the unconstrained global minimization problem: given a smooth nonconvex potential $U \colon \mathbb{R}^d \to \mathbb{R}$, we search for a point $x^*$ such that

$$x^* \in \operatorname{argmin}_{x \in \mathbb{R}^d} U(x),$$

assuming such a point exists.

A recent trend in optimization consists of studying continuous-time versions of algorithms to obtain better estimates for their discrete-time counterparts. Notable progress has been made in accelerating convergence of first-order optimization methods by analyzing second-order dynamical systems in their continuous-time formulation. In specific, we shall interest ourselves in the high-resolution differential equation, given by

$$\ddot{x}(t) + \alpha \dot{x}(t) + \beta \nabla^2 U(x(t))\dot{x}(t) + \gamma \nabla U(x(t)) = 0, \tag{1}$$

where $\alpha, \beta, \gamma > 0$ could in principle depend on time, but are supposed constants as indicated by the notation. The equation was originally introduced in Alvarez et al. (2002) and has since been further explored in Attouch et al. (2016), to mitigate the oscillations. The algorithmic consequences of this, including the

connection with Nesterov's method, were investigated independently in Attouch et al. (2022) and Shi et al. (2022). The theoretical convergence rates obtained for the resulting algorithmic framework were comparable to those established for Nesterov's method in the convex case. However, in the strongly convex case, the theoretical guarantees were more conservative. Nevertheless, this motivated intense research within the convex optimization community, and the high-resolution differential equation seems to be displacing the classical differential equation studied in Su et al. (2016) and Attouch et al. (2018), which corresponds to the overdamped Langevin system, as the preferred continuous-time model for Nesterov's acceleration. Amongst other reasons, the high-resolution model extends better to the nonsmooth setting, and it captures the linear convergence rates of FISTA and the Optimized Gradient Method in the strongly convex case, while the overdamped one does not. These characteristics, coupled with the more stable trajectories it enables and the potential for integrating other techniques, such as restart schemes (Su et al., 2016), underscore the high-resolution differential equation's significance as a promising area of interest in convex optimization. To the best of our knowledge, only the convergence to critical points has been shown for Equation (1) in nonconvex landscapes (Alvarez et al., 2002).

By setting $y(t) = \dot{x}(t) + \beta \nabla U(x(t))$ and renaming the parameter $\gamma$, we can rewrite Equation (1) as a first-order system

$$\begin{cases} \dot{x}(t) & = -\beta \nabla U(x(t)) + y(t) \\ \dot{y}(t) & = -\gamma \nabla U(x(t)) - \alpha y(t). \end{cases} \tag{2}$$

This reformulation has the benefit of not requiring the Hessian of $U$, making it more user-friendly, whilst still preserving the favourable convergence results.

However, deterministic models like System (2) may struggle when the potential $U$ is nonconvex, as they can become trapped in local minima and fail to identify the global minimizer. To overcome this limitation, it has been proposed to add stochasticity to the dynamics to enable them to escape local minima. This stochasticity can come in the form of random perturbations, encouraging the dynamics to navigate through more complex, potentially nonconvex, landscapes. This perspective naturally leads us to consider stochastic differential equations, specifically the Langevin dynamics (Langevin, 1908), which combine the advantages of gradient flows with stochastic elements. Rather than the iterates of these dynamics, we study their law, and hope for it to concentrate around the global minimizers of the potential. This pushes us towards sampling problems, where we aim to produce samples from a given distribution $\boldsymbol{\mu} \propto \exp(-U)$.

Many problems in statistics require sampling from probability distributions. Sampling through the Langevin dynamics is a well-studied approach when the target distribution is strongly log-concave (or equivalently, when the potential is strongly convex) (see Durmus and Moulines (2016); Dalalyan and Karagulyan (2019); Cheng and Bartlett (2018); Dalalyan and Riou-Durand (2020) and the references therein), and has recently also been studied in the non log-concave setting, when it, for instance, verifies a log-Sobolev inequality (Vempala and Wibisono, 2019; Ma et al., 2021), a Poincaré inequality (Chewi et al., 2022), a weak Poincaré inequality (Mousavi-Hosseini et al., 2023), or even in the fully nonconvex setting (Balasubramanian et al., 2022).

The simplest variant of the Langevin dynamics is the overdamped Langevin dynamics, governed by

$$dX_t = -\gamma \nabla U(X_t)dt + \sqrt{2\gamma}dB_t,$$

where $(B_t)$ is standard Brownian motion, $\gamma > 0$ is a free parameter and $U$ is the negative logarithm of the distribution we wish to sample from. Under weak assumptions, the invariant distribution of the dynamics is exactly $\boldsymbol{\mu} \propto \exp(-U)$. Convergence of the Euler discretization of the overdamped Langevin dynamics in Wasserstein-2 distance under strong convexity of $U$ was shown in Durmus and Moulines (2016), and convergence in Kullback-Leibler divergence under a log-Sobolev assumption on $\boldsymbol{\mu}$ in Vempala and Wibisono (2019). Accelerated rates may be obtained under different discretizations. For instance, see Dalalyan and Karagulyan (2019) for strongly log-concave targets.

A different approach to faster convergence involves the underdamped Langevin dynamics, governed by the stochastic differential equation

$$
\begin{cases}
dX_t & = V_t dt \\
dV_t & = (-\gamma \nabla U(X_t) - V_t)dt + \sqrt{2\gamma}dB_t,
\end{cases}
$$

where $V_t$ represents the velocity. Under weak assumptions, the invariant measure is $\boldsymbol{\mu}(x,v) \propto \exp(-U(x) - \|v\|^2/(2\gamma))$. Accelerated convergence with respect to the overdamped dynamics was shown in Wasserstein-2 distance in the strongly log-concave case by Cheng and Bartlett (2018) and in Kullback-Leibler divergence under a log-Sobolev inequality by Ma et al. (2021). We again note that different discretizations are possible, we cite for instance Dalalyan and Riou-Durand (2020) and Schuh and Whalley (2024).

Sampling algorithms for optimization were first introduced for strongly convex potentials in Dalalyan (2017), and further studied in Raginsky et al. (2017); Xu et al. (2018); Zhang et al. (2017); Tzen et al. (2018), in the nonconvex setting. All of these focused on discretizations of the overdamped Langevin dynamics. The underdamped Langevin dynamics have also been studied, we refer to Gao et al. (2020) and Borysenko and Byshkin (2021) for more details. Discretizations of other underlying processes were investigated in Chen et al. (2018) Zhang (2024). Langevin dynamics have also been studied for global optimization outside the context of sampling. We refer interested readers to Chen et al. (2024) and the references therein.

## 1.1 Contribution

The contributions of the paper are the following:

1. We design a global optimization algorithm capable of minimizing nonconvex functions and obtaining arbitrarily accurate solutions with high probability. This optimization algorithm relies on an oracle sampling algorithm.

2. We propose a new variant of the classical Langevin dynamics, both for continuous-time and discrete-time. The dynamics is inspired by the first-order high-resolution System (2), which is known to exhibit accelerated convergence rates in the deterministic convex setting. We anticipate that this study will encourage further exploration of this system, with the potential to deliver a comparable breakthrough in the field of nonconvex sampling as it has achieved in convex optimization. This sampling algorithm will then serve as oracle algorithm in our global optimization algorithm.

## 1.2 Structure

Section 2 introduces notation and preliminaries. Section 3 is dedicated to the design of our global optimization algorithm. In Section 4 we study a novel continuous- and discrete-time dynamics, and formulate the sampling results. Finally, in Section 5, we illustrate our results numerically on the Rastrigin function, where we improve on current methods. Technical results and proofs are provided in the appendix.

## 2 Notation and Assumptions

The following standing assumptions on the potential $U \colon \mathbb{R}^d \to \mathbb{R}$ will be valid throughout the paper:

- $U$ is twice differentiable, $L$-smooth and has Lipschitz continuous and bounded Hessian.

- There exists an $a_0 > 0$ such that $\exp(-a_0 U)$ is integrable.

- $U$ has a nonzero finite number of global minimizers, and admits no global minimizers at infinity. We define its minimal value to be $U^*$.

Let $\|\cdot\|$ denote the Euclidean norm on $\mathbb{R}^d$ and $\mathcal{P}(\mathbb{R}^d)$ denote the space of probability measures on $\mathbb{R}^d$. With abuse of notation, we shall denote interchangeably by $\boldsymbol{\mu} \in \mathcal{P}(\mathbb{R}^d)$ the probability distribution and its density function with respect to the Lebesgue measure, in the case where it exists. For a given potential $U \colon \mathbb{R}^d \to \mathbb{R}$ and reals $a \geq a_0$ and $b > 0$, we define $\boldsymbol{\mu}^a \in \mathcal{P}(\mathbb{R}^d)$ and $\boldsymbol{\mu}^{a,b} \in \mathcal{P}(\mathbb{R}^{2d})$ to be the probability distributions whose densities satisfy

$$
\boldsymbol{\mu}^a(x) \propto \exp(-aU(x)) \quad \text{and} \quad \boldsymbol{\mu}^{a,b}(x,y) \propto \exp\left(-aU(x) - b\frac{\|y\|^2}{2}\right),
$$

which are well-defined by assumption.

Given a distribution $\boldsymbol{\mu} \in \mathcal{P}(\mathbb{R}^d)$, we denote by

$$\mathbb{E}_{X \sim \boldsymbol{\mu}}[f(X)] = \int_{\mathbb{R}^d} f(x)\boldsymbol{\mu}(dx)$$

the expected value of $f(X)$ where $X \sim \boldsymbol{\mu}$. Whenever the random variable or its distribution is clear from context, we shall abbreviate this to $\mathbb{E}_{\boldsymbol{\mu}}[f(X)]$ or $\mathbb{E}[f(X)]$.

Let $\boldsymbol{\mu}, \boldsymbol{\nu} \in \mathcal{P}(\mathbb{R}^d)$ both have a density with respect to the Lebesgue measure and have full support. We define the **total variation distance** between $\boldsymbol{\mu}$ and $\boldsymbol{\nu}$ as

$$\|\boldsymbol{\mu} - \boldsymbol{\nu}\|_{\mathrm{TV}} = \sup\left\{|\mathbb{E}_{\boldsymbol{\mu}}[f] - \mathbb{E}_{\boldsymbol{\nu}}[f]| : \|f\|_\infty \le 1\right\}.$$

We define the **Kullback-Leibler divergence** of $\boldsymbol{\mu}$ with respect to $\boldsymbol{\nu}$ as

$$\mathrm{KL}(\boldsymbol{\mu}\|\boldsymbol{\nu}) = \mathbb{E}_{X \sim \boldsymbol{\mu}}\left[\log \frac{\boldsymbol{\mu}(X)}{\boldsymbol{\nu}(X)}\right].$$

Moreover, we define their **relative Fisher information** as

$$\mathrm{Fi}(\boldsymbol{\mu}\|\boldsymbol{\nu}) = \mathbb{E}_{X \sim \boldsymbol{\mu}}\left[\left\|\nabla \log \frac{\boldsymbol{\mu}(X)}{\boldsymbol{\nu}(X)}\right\|^2\right].$$

Finally, we define the **Wasserstein-$2$ distance** between $\boldsymbol{\mu}$ and $\boldsymbol{\nu}$ as

$$W_2(\boldsymbol{\mu}, \boldsymbol{\nu}) = \left(\inf_{\boldsymbol{\zeta} \in \Gamma(\boldsymbol{\mu}, \boldsymbol{\nu})} \mathbb{E}_{(X,Y) \sim \boldsymbol{\zeta}}\left[\|X - Y\|_2^2\right]\right)^{1/2},$$

where $\Gamma(\boldsymbol{\mu}, \boldsymbol{\nu})$ is the set of couplings of $\boldsymbol{\mu}$ and $\boldsymbol{\nu}$, namely the set of distributions $\boldsymbol{\zeta} \in \mathcal{P}(\mathbb{R}^{2d})$ such that $\boldsymbol{\zeta}(A \times \mathbb{R}^d) = \boldsymbol{\mu}(A)$ and $\boldsymbol{\zeta}(\mathbb{R}^d \times A) = \boldsymbol{\nu}(A)$ for all $A \in \mathcal{B}(\mathbb{R}^d)$. The infimum is always attained, and we call the minimizers optimal couplings (Villani, 2009).

A standard assumption in the literature when dealing with nonconvex sampling is a logarithmic Sobolev inequality (Vempala and Wibisono, 2019; Ma et al., 2019; 2021), which may be viewed as a Polyak-Łojasiewicz inequality on the space of probability measures (Liu et al., 2023; Chewi and Stromme, 2024). To the best of our knowledge, this assumption is not standard in the field of global optimization. A log-Sobolev inequality on $\boldsymbol{\mu}$ with coefficient $\rho$ states that, for all $\boldsymbol{\nu} \in \mathcal{P}(\mathbb{R}^d)$,

$$\mathrm{KL}(\boldsymbol{\nu}\|\boldsymbol{\mu}) \le \frac{1}{2\rho}\mathrm{Fi}(\boldsymbol{\nu}\|\boldsymbol{\mu}). \tag{3}$$

*Remark* 2.1. The notion of a logarithmic Sobolev inequality as presented in Inequality (3) was originally introduced in Feissner (1972) for Gaussian measures, and extended in Gross (1975) for general measures. All strongly log-concave probability distributions satisfy the inequality (Bakry and Émery, 1985), which is stable under bounded perturbations (Holley and Stroock, 1987), tensorization, convolution and mixture (Ledoux, 2006; Bakry et al., 2014), and Lipschitz perturbations (Brigati and Pedrotti, 2024), although this might come at the expense of the constant. In the case of mixtures of Gaussians of equal variance, the log-Sobolev constant has an exponential dependency in the problem dimension (Menz and Schlichting, 2014; Schlichting, 2019). Moreover, measures with potentials which are strongly convex outside a ball satisfy a log-Sobolev inequality (Ma et al., 2019), although the constant may again have an exponential dependency in the dimension.

**Assumption 2.2.** The Gibbs measures $\boldsymbol{\mu}^a$ of $U: \mathbb{R}^d \to \mathbb{R}$ satisfy a logarithmic Sobolev inequality with coefficients $\rho_a > 0$, for all $a \ge a_0$.

*Remark* 2.3. Under Assumption 2.2, the measures $\boldsymbol{\mu}^{a,b}$ also satisfy a log-Sobolev inequality with constant $\rho_{a,b} = \min(\rho_a, \rho_b)$ by tensorization, where we know that $\rho_b = 1/b$ since the marginal in $y$ of $\boldsymbol{\mu}^{a,b}$ is strongly log-concave with parameter $b$. Verifying the logarithmic Sobolev inequality for each $a \geq a_0$ may be challenging in general, however it is readily verified under some structural assumptions on the potential. For instance, it is satisfied if $U = V + F$, where $V$ is strongly convex and $F$ is bounded, as strongly convex potentials induce a Gibbs measure satisfying the log-Sobolev property, which is stable under bounded perturbations. To the best of our knowledge, the tightest known lower bound for $\rho_a$ is given by $\exp(-a \operatorname{Osc}(F)) \cdot (a\mu)^{-1}$, where $\mu$ is the strong convexity constant of $V$. As $\operatorname{Osc}(F)$ typically scales with the dimension $d$, this highlights the possible exponential dependency in the dimension and in the parameter $a$ of $\rho_a$, and hence also of $\rho_{a,b}$.

We now recall Pinsker's Inequality (Pinsker, 1964), which relates the KL divergence and the total variation distance.

**Theorem 2.4** (Pinsker's Inequality). *For any two $\boldsymbol{\mu}, \boldsymbol{\nu} \in \mathcal{P}(\mathbb{R}^d)$, it holds that*

$$\|\boldsymbol{\mu} - \boldsymbol{\nu}\|_{\mathrm{TV}} \leq \sqrt{\frac{\mathrm{KL}(\boldsymbol{\nu}\|\boldsymbol{\mu})}{2}}.$$

We finish this section by recalling McDiarmid's Inequality (McDiarmid, 1989) in the single-variable case.

**Lemma 2.5.** *Let $f\colon \mathbb{R}^d \to \mathbb{R}$ satisfy $\operatorname{Osc}(f) \coloneqq \sup(f) - \inf(f) < +\infty$. For any random variable $X$, it holds that*

$$\mathbb{P}(\mathbb{E}[f(X)] - f(X) \geq \varepsilon) \leq \exp\left(-\frac{2\varepsilon^2}{\operatorname{Osc}(f)^2}\right).$$

## 3 Optimization

The idea behind the global optimization algorithm is to sample from a distribution that produces samples close to global minimizers. Specifically, we define $\boldsymbol{\mu}^* \in \mathcal{P}(\mathbb{R}^d)$ to be a mixture of Dirac measures concentrated around the global minimizers of $U$ with weights as defined in (Hasenpflug et al., 2024, Equation 18). As such, by Hasenpflug et al. (2024), under the technical Assumption B.1, there exists a constant $C > 0$, depending on $U$ only, satisfying

$$W_2(\boldsymbol{\mu}^a, \boldsymbol{\mu}^*) \leq C \cdot a^{-1/4}. \tag{4}$$

As such, an approximate sample from $\boldsymbol{\mu}^a$ will yield samples close to global minimizers. Concretely, we suppose we have access to a distribution $\tilde{\boldsymbol{\mu}}$ satisfying

$$\mathrm{KL}(\tilde{\boldsymbol{\mu}}\|\boldsymbol{\mu}^a) \leq \varepsilon^2/18, \tag{5}$$

for some $\varepsilon > 0$, and that we can draw samples from $\tilde{\boldsymbol{\mu}}$.

We are now ready to introduce our global optimization algorithm, dependent on a yet unspecified oracle sub-algorithm, corresponding to a sample from $\tilde{\boldsymbol{\mu}}$.

---

**Algorithm 1** Global Optimization Algorithm

**Require:** Oracle algorithm.
 1: Generate $N$ random i.i.d. samples $\tilde{X}^{(i)}$ according to oracle algorithm where $i = 1, \dots, N$.
 2: Set $\tilde{X} = \tilde{X}^{(I)}$ for $I = \operatorname{argmin}_{i=1\dots,N} U(\tilde{X}^{(i)})$.

---

With these results at hand, we may prove convergence in probability of Algorithm 1.

**Theorem 3.1.** *Let $U\colon \mathbb{R}^d \to \mathbb{R}$ satisfy Assumptions 2.2 and B.1. Fix $\varepsilon \in (0, 1/2)$, $\delta \in (0, 1)$, and suppose*

$$a \geq \max\left(a_0, \frac{9C^4 L^2}{\varepsilon^2}\right) \quad and \quad N \geq \frac{18 \ln(1/\delta)}{\varepsilon^2}. \tag{6}$$

*Suppose we have access to an oracle algorithm that can produce a sample from $\tilde{\boldsymbol{\mu}}$, where $\tilde{\boldsymbol{\mu}}$ satisfies (5). Then, if $\tilde{X}$ is simulated according to Algorithm 1 with the same oracle algorithm, it holds that*

$$\mathbb{P}(U(\tilde{X}) - U^* \geq \varepsilon) \leq \delta.$$

*Remark* 3.2. A natural choice for the oracle algorithm producing samples $\tilde{X}^{(i)}$ satisfying (5) is an iterative sampling algorithm, producing the sample in $K$ iterations. The total complexity to produce the sample $\tilde{X}$ is then of $K \cdot N$, with the possibility of parallelization when $N > 1$. For a fixed computational budget, there is a trade-off between the number of iterations $K$ and the number $N$ of samples, as indicated by Equation (8) below, where the number of iterations $K$ will control how small the probability $\mathbb{P}(U(\tilde{X}^{(1)}) \geq \varepsilon)$ is. Therefore, the number of iterations $K$ should be big enough if one wants to ensure that increasing the number of samples $N$ leads to improved accuracy.

*Proof.* Without loss of generality, set $U^* = 0$.

Let $\tilde{X}^{(i)}$ for $i = 1, \ldots, N$ be $N$ i.i.d. copies generated according to the oracle algorithm, such that $\tilde{X} = \tilde{X}^{(I)}$ where $I = \operatorname{argmin}_{i=1,\ldots,N} U(\tilde{X}^{(i)})$.

Note that, for any $i = 1, \ldots, N$,

$$1 - e^{-U(\tilde{X}^{(i)})} = 1 - e^{-\mathbb{E}[U(X^a)]} \tag{7a}$$

$$+ e^{-\mathbb{E}[U(X^a)]} - \mathbb{E}[e^{-U(X^a)}] \tag{7b}$$

$$+ \mathbb{E}[e^{-U(X^a)}] - \mathbb{E}[e^{-U(\tilde{X}^{(i)})}] \tag{7c}$$

$$+ \mathbb{E}[e^{-U(\tilde{X}^{(i)})}] - e^{-U(\tilde{X}^{(i)})}. \tag{7d}$$

Term (7a) is bounded by $L$-smoothness as

$$\mathbb{E}[U(X^a)] = \mathbb{E}[U(X^a) - U(X^*)]$$

$$\leq \mathbb{E}\left[\nabla U(X^*)(X^a - X^*) + \frac{L}{2}\|X^a - X^*\|^2\right]$$

$$= \frac{L}{2}W_2^2(\boldsymbol{\mu}^a, \boldsymbol{\mu}^*) \leq \frac{LC^2}{2\sqrt{a}} \leq \frac{\varepsilon}{6},$$

where $X^* \sim \boldsymbol{\mu}^*$ such that $\nabla U(X^*) = 0$ almost surely, and $(X^a, X^*) \sim \boldsymbol{\zeta}^*$ for $\boldsymbol{\zeta}^*$ an optimal coupling between $\boldsymbol{\mu}^a$ and $\boldsymbol{\mu}^*$. As such, using that $1 - e^{-x} \leq x$, we obtain that (7a) is bounded by $\varepsilon/6$. Moreover, as $x \mapsto \exp(-x)$ is convex, (7b) is nonpositive, by Jensen's inequality. Since $x \mapsto \exp(-U(x))$ is bounded by 1, (7c) is bounded as

$$\mathbb{E}\left[e^{-U(X^a)}\right] - \mathbb{E}\left[e^{-U(\tilde{X}^{(0)})}\right] \leq \|\tilde{\boldsymbol{\mu}} - \boldsymbol{\mu}^a\|_{TV} \leq \sqrt{\frac{\mathrm{KL}(\tilde{\boldsymbol{\mu}}\|\boldsymbol{\mu}^a)}{2}} \leq \frac{\varepsilon}{6},$$

where the second step uses Pinsker's Inequality 2.4. Finally, by considering all the bounds on (7), we have

$$\mathbb{P}\left(1 - e^{-U(\tilde{X}^{(i)})} \geq \varepsilon/2\right) \leq \mathbb{P}\left(\mathbb{E}[e^{-U(\tilde{X}^{(i)})}] - e^{-U(\tilde{X}^{(i)})} \geq \varepsilon/6\right) \leq \exp(-\varepsilon^2/18) \leq \delta^{1/N},$$

where the third inequality follows by Lemma 2.5, as $x \mapsto e^{-U(x)}$ takes values in $[0, 1]$. For $x \in [0, 1/2]$, it holds that $1 - e^{-x} \geq x/2$, and hence

$$\mathbb{P}(U(\tilde{X}^{(i)}) \geq \varepsilon) \leq \mathbb{P}\left(1 - e^{-U(\tilde{X}^{(i)})} \geq \varepsilon/2\right) \leq \delta^{1/N}.$$

As $\tilde{X} = \tilde{X}^{(I)}$ where $I = \operatorname{argmin}_{i=1,\ldots,N} U(\tilde{X}^{(i)})$, and $(\tilde{X}^{(i)})_{i=1,\ldots,N}$ are i.i.d., it holds that

$$\mathbb{P}(U(\tilde{X}) \geq \varepsilon) = \mathbb{P}(U(\tilde{X}^{(1)}) \geq \varepsilon)^N \leq \delta. \tag{8}$$

$\square$

*Remark* 3.3. In Theorem 3.1, the bound of $\varepsilon < 1/2$ is artificial to the proof. By scaling $U$ one can obtain similar results, up to a constant factor, for any value of $\varepsilon > 0$.

Algorithm 1 depends on a yet unspecified oracle algorithm. A new such oracle algorithm is discussed in Section 4. However, the modular form of Algorithm 1 provides a simple framework to employ other sampling algorithms, which potentially may lead to faster provable convergence, or be applicable in different contexts.

## 4  High-Resolution Langevin

### 4.1  Continuous-Time Study

As previously highlighted, the high-resolution differential equation introduced in System (2) has played a pivotal role in advancing the study of accelerated convex optimization algorithms. Aiming to build an algorithm with accelerated convergence rate for nonconvex sampling, we introduce the *High-Resolution Langevin Dynamics*, inspired by these dynamics. Let $(\Omega, \mathcal{F}, \mathbb{P})$ be a filtered probability space and consider the following stochastic differential equation:

$$
\begin{cases}
dX_t = (-\beta \nabla U(X_t) + Y_t)dt + \sqrt{2\sigma_x^2}dB_t^x \\
dY_t = (-\gamma \nabla U(X_t) - \alpha Y_t)dt + \sqrt{2\sigma_y^2}dB_t^y,
\end{cases}
\tag{9}
$$

where $(B^x, B^y)$ is a standard $2d$-dimensional Brownian motion, $(X_0, Y_0) \sim \boldsymbol{\mu}_0$ for some initial distribution $\boldsymbol{\mu}_0$, and $\alpha, \beta, \gamma, \sigma_x^2, \sigma_y^2 > 0$. As the drift coefficient is Lipschitz continuous, System (9) has a unique solution (Friedman, 1975). We denote the joint law of $(X_t, Y_t)$ by $\boldsymbol{\mu}_t$, which has a twice continuously differentiable density with respect to the Lebesgue measure, as the drift coefficient has Lipschitz continuous and bounded gradient (Menozzi et al., 2021).

A slight variation of System (9) has been studied previously in Li et al. (2022), with similar motivations. The authors obtain convergence rates in the $W_2$ metric, in the strongly log-concave setting. As our study addresses the nonconvex setting, a comparative analysis falls outside the scope of this work.

*Remark* 4.1. System (9) is equivalent to

$$
dZ_t = -\begin{pmatrix} \beta/a & -1/b \\ \gamma/a & \alpha/b \end{pmatrix} \nabla H(Z_t)dt + 2\begin{pmatrix} \sigma_x^2 & 0 \\ 0 & \sigma_y^2 \end{pmatrix} dB_t,
$$

where $Z_t = (X_t, Y_t)$ and $H((x,y)) = aU(x) + b\frac{\|y\|^2}{2}$. As such, System (9) may be viewed as a preconditioned Langevin dynamics in a larger space on the function $H$.

Moreover, we see the difference between System (9) and the underdamped Langevin dynamics, through the presence of additional noise.

Finally, as $\sigma_x^2, \sigma_y^2 \to 0$, we recover the deterministic System (2), which exhibits accelerated convergence.

We now study the convergence in KL divergence of System (9), similarly to what was done in Ma et al. (2021). A byproduct of the proof (see Appendix C.1) is the uniqueness of the invariant measure.

**Theorem 4.2.** *Let $U \colon \mathbb{R}^d \to \mathbb{R}$, and let $a \geq a_0$ and $b, \alpha, \beta, \gamma, \sigma_x, \sigma_y > 0$ satisfy*

$$
a = \frac{\beta}{\sigma_x^2}, \quad b = \frac{\alpha}{\sigma_y^2} \quad and \quad \frac{a}{b} = \gamma.
\tag{10}
$$

*1. System (9) admits a weak solution $(X_t, Y_t)$ which has as invariant law $\boldsymbol{\mu}^{a,b}$.*

*2. If $\boldsymbol{\mu}^{a,b}$ satisfies a log-Sobolev inequality with $\rho > 0$,*

$$
\mathrm{KL}(\boldsymbol{\mu}_t \| \boldsymbol{\mu}^{a,b}) \leq \mathrm{KL}(\boldsymbol{\mu}_0 \| \boldsymbol{\mu}^{a,b}) \cdot e^{-2\rho \min(\sigma_x^2, \sigma_y^2)t}.
$$

*In specific, provided $\sigma_x^2, \sigma_y^2 > 0$, we obtain $\mathrm{KL}(\boldsymbol{\mu}_t \| \boldsymbol{\mu}^{a,b}) \to 0$ at exponential rate as $t \to \infty$, and the invariant law $\boldsymbol{\mu}^{a,b}$ is unique.*

### 4.2  Discrete-Time Study

Consider the following discretization of System (9):

$$
\begin{cases}
d\tilde{X}_t = (-\beta \nabla U(\tilde{X}_{kh}) + \tilde{Y}_t)dt + \sqrt{2\sigma_x^2}dB_t^x \\
d\tilde{Y}_t = (-\gamma \nabla U(\tilde{X}_{kh}) - \alpha \tilde{Y}_t)dt + \sqrt{2\sigma_y^2}dB_t^y,
\end{cases}
\tag{11}
$$

for $t \in [kh, (k+1)h]$, where $h > 0$ is the step-size and $\tilde{\boldsymbol{\mu}}_0$ is the initial distribution, such that $(\tilde{X}_0, \tilde{Y}_0) \sim \tilde{\boldsymbol{\mu}}_0$. We define $\tilde{\boldsymbol{\mu}}_t = \mathcal{L}((\tilde{X}_t, \tilde{Y}_t))$.

Conditionally on $(\tilde{X}_{kh}, \tilde{Y}_{kh})$, System (11) describes an Ornstein-Uhlenbeck process for $t \in [kh, (k+1)h]$. We may thus simulate $\tilde{\boldsymbol{\mu}}_{(k+1)h}$ by sampling an appropriate Gaussian random variable. The explicit computations leading to Algorithm 2 are presented in Appendix C.2.

---

**Algorithm 2** High-Resolution Sampling Algorithm

---

**Require:** An initial distribution $\tilde{\boldsymbol{\mu}}_0 \in \mathcal{P}(\mathbb{R}^{2d})$.
1: Simulate $(\tilde{X}_0, \tilde{Y}_0) \sim \tilde{\boldsymbol{\mu}}_0$.
2: **for** $k = 0, \ldots, K-1$ **do**
3:     Generate $(\tilde{X}_{(k+1)h}, \tilde{Y}_{(k+1)h}) \sim \mathcal{N}(m, \Sigma)$ conditionally on $(\tilde{X}_{kh}, \tilde{Y}_{kh})$, where

$$
\begin{cases}
m_X = \tilde{X}_{kh} - \beta h \nabla U(\tilde{X}_{kh}) + \dfrac{1 - e^{-\alpha h}}{\alpha} \tilde{Y}_{kh} - \dfrac{\gamma}{\alpha}\left(h - \dfrac{1 - e^{-\alpha h}}{\alpha}\right)\nabla U(\tilde{X}_{kh}) \\[2mm]
m_Y = e^{-\alpha h} \tilde{Y}_{kh} - \dfrac{\gamma}{\alpha}(1 - e^{-\alpha h})\nabla U(\tilde{X}_{kh}) \\[2mm]
\Sigma_{XX} = \dfrac{\sigma_y^2}{\alpha^3}\left[2\alpha h - e^{-2\alpha h} + 4e^{-\alpha h} - 3\right] \cdot I_d + 2\sigma_x^2 h \cdot I_d \\[2mm]
\Sigma_{YY} = \dfrac{\sigma_y^2(1 - e^{-2\alpha h})}{\alpha} \cdot I_d \\[2mm]
\Sigma_{XY} = \Sigma_{YX} = \dfrac{\sigma_y^2(1 - e^{-\alpha h})^2}{\alpha^2} \cdot I_d.
\end{cases}
$$

4: **end for**
5: **return** $(\tilde{X}_{Kh}, \tilde{Y}_{Kh})$.

---

Our result, as well as our analysis, is comparable to Vempala and Wibisono (2019), who studied the highly overdamped Langevin Dynamics. The more precise result and its derivation are given in Appendix C.3.

**Theorem 4.3.** *Let $\varepsilon > 0$, $a \geq a_0$, $b > 0$ and assume (10) holds. If a log-Sobolev inequality on $\boldsymbol{\mu}^{a,b}$ with parameter $\rho > 0$ holds, and $h \lesssim \mathcal{O}(\rho)$, then there exist*

$$\tilde{A} = \mathcal{O}(\rho), \quad \tilde{B} = \mathcal{O}(a^2 d/\rho) \quad and \quad \hat{B} = \mathcal{O}(a^2 d),$$

*such that*

$$\mathrm{KL}(\tilde{\boldsymbol{\mu}}_h \| \boldsymbol{\mu}^{a,b}) \leq e^{-\tilde{A}h}\,\mathrm{KL}(\tilde{\boldsymbol{\mu}}_0 \| \boldsymbol{\mu}^{a,b}) + \hat{B}h^2,$$

*and, for all $K \geq 1$,*

$$\mathrm{KL}(\tilde{\boldsymbol{\mu}}_{Kh} \| \boldsymbol{\mu}^{a,b}) \leq e^{-\tilde{A}Kh}\,\mathrm{KL}(\tilde{\boldsymbol{\mu}}_0 \| \boldsymbol{\mu}^{a,b}) + \tilde{B}h. \tag{12}$$

As an immediate corollary we obtain sufficient conditions to obtain an $\varepsilon$-accurate sample.

**Corollary 4.4.** *Let $\varepsilon > 0$, $a \geq a_0$, $b > 0$ and assume (10) holds. If a log-Sobolev inequality on $\boldsymbol{\mu}^{a,b}$ with parameter $\rho > 0$ holds, then $\mathrm{KL}(\tilde{\boldsymbol{\mu}}_{Kh} \| \boldsymbol{\mu}^{a,b}) \leq \varepsilon$ for*

$$h \lesssim \tilde{\mathcal{O}}\left(\frac{\rho \varepsilon}{a^2 d}\right) \quad and \quad K \gtrsim \tilde{\mathcal{O}}\left(\frac{d a^2}{\rho^2 \varepsilon}\right).$$

*Proof.* Bound each term in Equation (12) by $\varepsilon/2$. $\qquad\square$

We are now ready to complement Theorem 3.1, by replacing the oracle algorithm by Algorithm 2.

**Corollary 4.5.** *Let* $U \colon \mathbb{R}^d \to \mathbb{R}$ *satisfy Assumptions 2.2 and B.1, and fix* $\varepsilon \in (0, 1/2)$ *and* $\delta \in (0, 1)$*. Suppose* (6) *holds, and that* $\tilde{X}$ *is simulated according to Algorithm 1 with $K$ iterations of Algorithm 2 being used for the oracle sub-procedure. It holds that*

$$\mathbb{P}(U(\tilde{X}) - U^* \geq \varepsilon) \leq \delta, \quad for \quad h \lesssim \tilde{\mathcal{O}}\left(\frac{\rho_{a,b}\varepsilon^2}{a^2 d}\right) \quad and \quad K \gtrsim \tilde{\mathcal{O}}\left(\frac{da^2}{\varepsilon^2 \rho_{a,b}^2}\right).$$

*Remark* 4.6. As in Remark 3.3, the results in Corollary 4.5 immediately extend to any $\varepsilon > 0$.

*Remark* 4.7. It may be observed that the rates presented in Corollary 4.5 do not improve upon the best-known rates for the overdamped Langevin dynamics (Vempala and Wibisono, 2019) or the underdamped Langevin dynamics (Ma et al., 2021). However, this does not imply the absence of acceleration; rather, it reflects that our current analytical framework is not sufficiently tight to capture it. This is comparable to the initial studies on high-resolution differential equations which similarly did not demonstrate provable acceleration, yet laid the groundwork for the remarkable advancements in acceleration techniques that we benefit from today.

## 5 Numerical Results

All experiments have been performed in Python 3.8. The code is available on the author's GitHub page.[1] Our theoretical study does not answer the question of the optimality of the parameter choice, which we leave for future work. Given $a > 0$, we fix $\alpha = 1$, $\beta = 1$, $b = 10$, $\gamma = a/10$, $\sigma_x^2 = 1/a$ and $\sigma_y^2 = 0.1$. The remaining parameters, namely the number of samples $N$, the number of iterations $K$ and the step-size $h$, will vary with the experiments. The number of runs over which we compute empirical probabilities is denoted by $M$.

We illustrate the convergence of our algorithm on the Rastrigin function, a classical example of a highly multimodal function with regularly distributed local minima. Let $U \colon \mathbb{R}^d \to \mathbb{R}$ be given by

$$U(x) = d + \|x\|^2 - \sum_{i=1}^d \cos(2\pi x_i),$$

which is minimized in $x^* = (0, \ldots, 0) \in \mathbb{R}^d$, with objective value $U^* = U(x^*) = 0$. The Rastrigin function for $d = 1$ and $d = 2$ is plotted in Figure 6 of Appendix A for illustrative purposes. The Gibbs measure of the Rastrigin function satisfies a log-Sobolev inequality by Remark 2.3, and it is easy to see it also satisfies Assumption B.1. We select $d = 10$ for all the experiments, unless otherwise specified.

In Figure 1, we show the empirical probabilities computed over $M = 100$ runs that $U(\tilde{X}_k) - U^* \geq \varepsilon$, for various thresholds $\varepsilon$. In each run, a step-size $h = 0.01$, a sample number $N = 10$ and a maximal number of iterations $K = 14000$ have been chosen. The initial distribution is set to $\tilde{\boldsymbol{\mu}}_0 = \mathcal{N}(3 \cdot \mathbf{1}_d, 10 \cdot I_{d \times d})$. We observe that for smaller values of $a$, $\boldsymbol{\mu}^a$ is not representative enough of $\boldsymbol{\mu}^*$ to guarantee the wanted threshold, even after numerous iterations. For larger $a$, the probability converges, with a rate that decreases as $a$ increases. This is expected, as $\boldsymbol{\mu}^a$ approaches $\boldsymbol{\mu}^*$ as $a$ increases, but the number of iterations to reach a good estimate of $\boldsymbol{\mu}^a$ also increases as $a$ increases. These observations qualitatively confirm Corollary 4.5, as well as the dependency in $a$ of $\rho_{a,b}$ as outlined in Remark 2.3.

Table 1 shows similar results in a tabular form, for various values of the step-size $h$. We report the average, median and standard deviation of $M = 100$ runs after $K = 14000$ iterations. The small standard deviation showcases the robustness of our method. These results motivate our selection of step-size for the subsequent experiments.

As previously noted, Algorithm 1 achieves convergence for any sampling algorithm that satisfies the conditions outlined in Theorem 3.1. In particular, discretizations of both overdamped and underdamped Langevin dynamics are anticipated to be viable candidates. In Figure 2, we compare the high-resolution Langevin algorithm (HRLA) proposed in Algorithm 2 with the overdamped Langevin algorithm (OLA) from Vempala and Wibisono (2019) and the underdamped Langevin algorithm (ULA) from Ma et al. (2021), using comparable parameter settings. The values of $a$ are selected empirically to optimize the convergence rate for a given value

---

[1]https://github.com/DanielCortild/GlobalOptimization

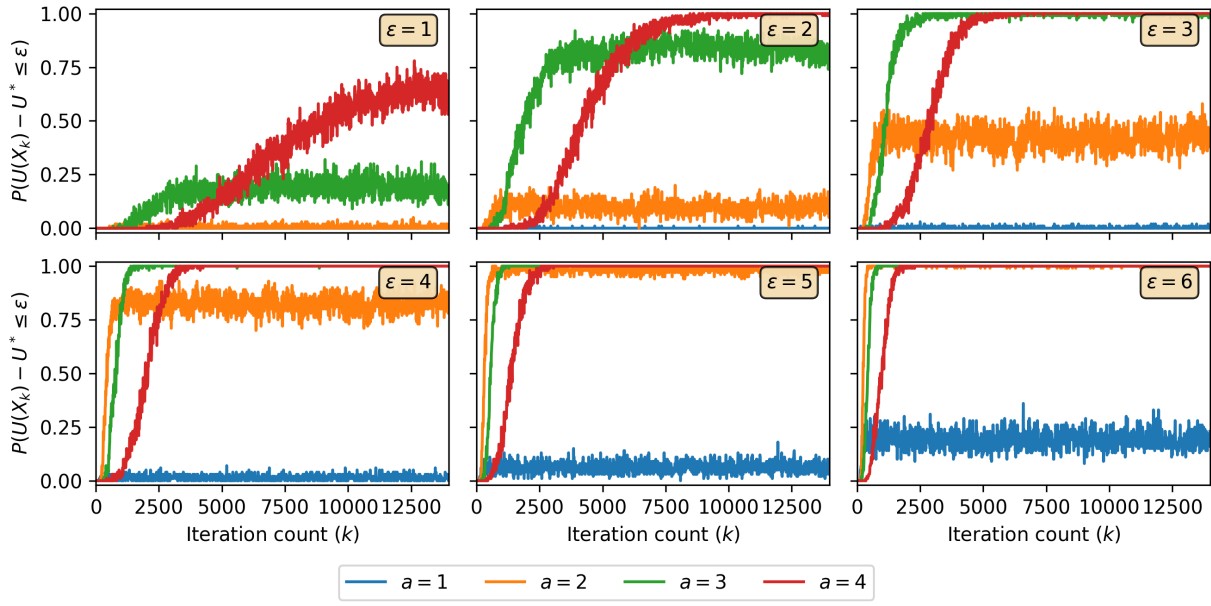

Figure 1: Empirical Probabilities of Hitting Tolerance $\varepsilon$ for 14000 Iterations.

Table 1: Average, Median and Standard Deviation of Objective Function Value for Various Step-Sizes.

| $h$ | | $a = 1$ | $a = 2$ | $a = 3$ | $a = 4$ |
|---|---|---|---|---|---|
| 0.001 | Avg | 2.974 | 0.920 | 1.363 | 4.216 |
| 0.001 | Med | 3.034 | 0.895 | 1.427 | 4.291 |
| 0.001 | SD | 0.595 | 0.334 | 0.617 | 0.906 |
| 0.01 | Avg | 3.422 | 0.949 | 0.425 | 0.318 |
| 0.01 | Med | 3.488 | 0.960 | 0.422 | 0.329 |
| 0.01 | SD | 0.599 | 0.257 | 0.101 | 0.076 |
| 0.1 | Avg | 7.840 | 6.970 | 6.780 | 6.570 |
| 0.1 | Med | 7.977 | 7.168 | 6.894 | 6.712 |
| 0.1 | SD | 1.031 | 0.959 | 0.842 | 0.766 |

of $\varepsilon$, with convergence assessed by the speed at which the empirical probability reaches 1, which reflects the accuracy of the invariant measure and does not depend on the sampling algorithm. Hence the chosen pairs $(a, \varepsilon)$ are common to all three algorithms.

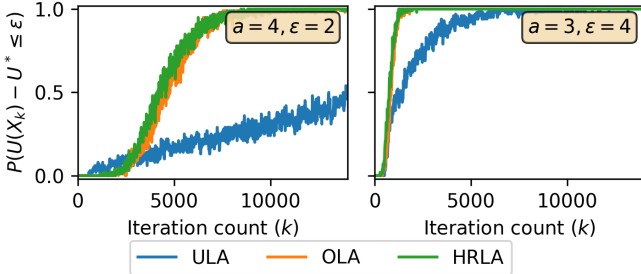

Figure 2: Comparison of Different Sampling Methods.

Extending the results of Figure 1 to a larger number of iterations ($K = 100000$) with larger values of $a$ allows us to reach better accuracies. This is shown in Figure 3, in which we observe the same trends as in Figure 1.

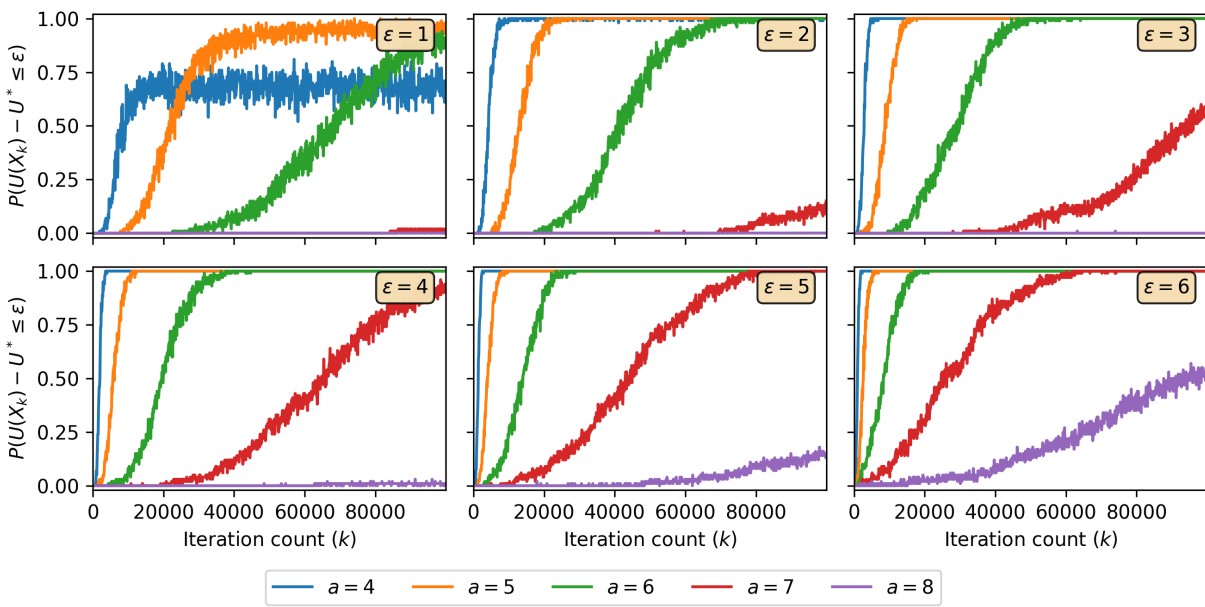

Figure 3: Empirical Probabilities of Hitting Tolerance $\varepsilon$ for 100000 Iterations.

The impact of the problem's dimensionality can also be analyzed. In Figure 4, we compare the cases of $d = 10$ with $\varepsilon = 2$ and $d = 20$ with $\varepsilon = 4$. The observed trends are remarkably similar, which aligns with theoretical expectations: doubling the dimension proportionally the expected initial objective value, and consequently, doubling the tolerance maintains the relative error at a consistent level.

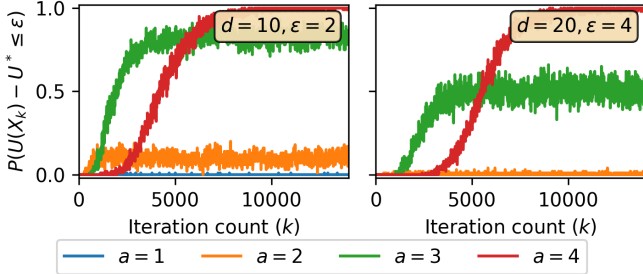

Figure 4: Empirical Probabilities of Hitting Tolerance $\varepsilon$ for $d = 10$ and $d = 20$.

In all preceding figures, we observe a faster convergence to the target $\boldsymbol{\mu}^a$ for smaller $a$, however the target $\boldsymbol{\mu}^a$ is further from the true target $\boldsymbol{\mu}^*$, such that the probability converges to a value far from 1. In order to leverage this, one can update the value of $a$ over the iterations, in the same spirit as *simulated annealing* (Gidas, 1985). For a given $\underline{a}$ and $\overline{a}$, we let $a_k$ be the value of $a$ at iteration $k$, where we assume $(a_k)$ evolves linearly between $\underline{a}$ and $\overline{a}$ in $k$. Specifically, we set

$$a_k = \frac{(K - k) \cdot \underline{a} - k \cdot \overline{a}}{K}, \tag{13}$$

where $K$ is the total number of iterations. We select $\underline{a} = 0.1$, and vary the final value $\overline{a}$. Figure 5 plots the empirical probabilities for various values of $\overline{a}$, now for smaller tolerances. We observe a faster convergence to a higher accuracy, as expected. For large values of $\overline{a}$, the increase from $\underline{a}$ to $\overline{a}$ is abrupt, causing the algorithm to become trapped in local minima. This behavior may originate from the suboptimality of the cooling scheme described in Equation (13). Further acceleration may be obtained either by optimizing the

Table 2: Average Objective Function Value for Fixed Effort $N \cdot K = 140000$, with $a_k$ given by Equation (13).

| $N$ | $\bar{a} = 4$ | $\bar{a} = 12$ | $\bar{a} = 20$ | $\bar{a} = 40$ |
|---|---|---|---|---|
| 1 | **0.140** | 0.062 | 0.078 | **0.141** |
| 10 | 0.143 | **0.048** | **0.054** | 0.176 |
| 100 | 0.197 | 0.229 | 1.311 | 4.432 |
| 1000 | 3.653 | 8.781 | 11.167 | 12.558 |
| 10000 | 16.206 | 16.014 | 16.023 | 16.160 |

Table 3: Comparison to Results in Guilmeau et al. (2021), with $a_k$ given by Equation (13).

| $K$ | | SA | FSA | SMC-SA | CSA | $\bar{a} = 1$ | $\bar{a} = 2$ | $\bar{a} = 3$ | $\bar{a} = 4$ | $\bar{a} = 5$ | $\bar{a} = 6$ |
|---|---|---|---|---|---|---|---|---|---|---|---|
| 50 | Avg | 3.29 | 3.36 | 3.26 | **3.23** | 15.76 | 15.30 | 14.04 | 13.61 | 13.40 | 13.40 |
| 50 | SD | **0.425** | 0.453 | 0.521 | 0.484 | 2.539 | 2.262 | 2.563 | 2.068 | 2.306 | 2.065 |
| 500 | Avg | 2.52 | 2.64 | 2.62 | 2.47 | 2.56 | 0.74 | 0.38 | 0.32 | **0.31** | 0.61 |
| 500 | SD | 0.320 | 0.304 | 0.413 | 0.502 | 0.549 | 0.244 | 0.101 | **0.095** | 0.223 | 0.433 |

cooling scheme, or by employing alternatives to simulated annealing. Although we do not delve further into this, we refer the interested reader to Marinari and Parisi (1992).

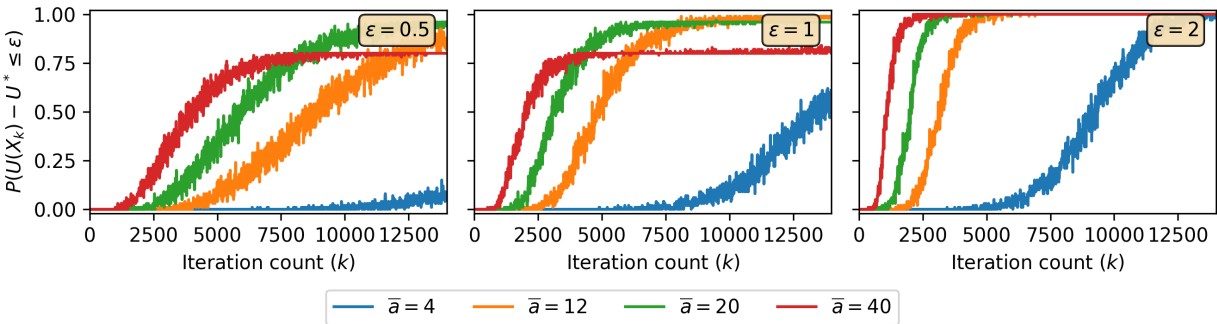

Figure 5: Empirical Probabilities of Hitting Tolerance $\varepsilon$ with $a_k$ given by Equation (13), for 14000 Iterations.

For a fixed computational effort $N \cdot K$, one can choose to put a heavier emphasis on the number of samples $N$ or the number of iterations $K$. Table 2 shows a comparison between multiple pairs of $(N, K)$ having a constant product and the average running best value over all iterates obtained. We observe that it is beneficial to select $N$ small, although $N = 1$ is not always optimal. Moreover, $N = 1$ does not allow for parallelization, contrary to $N > 1$. This does align with the observations in Remark 3.2.

In Table 3 we compare our results to well-studied variants of simulated annealing. Our algorithm is compared to *Simulated Annealing* (SA) as presented in Haario and Saksman (1991), *Fast Simulated Annealing* (FSA) as presented in Rubenthaler et al. (2009), *Sequential Monte Carlo Simulated Annealing* (SMC-SA) as presented in Zhou and Chen (2013), and *Curious Simulated Annealing* as presented in Guilmeau et al. (2021). The numerical values for the four aforementioned methods are extracted from Guilmeau et al. (2021). For a fair comparison we use the same parameters, namely we perform $M = 50$ runs with $N = 250$ samples and $K = 500$ number of iterations, and an initial distribution $\tilde{\boldsymbol{\mu}}_0 = \delta_{\{x_0\}}$, where $x_0 = (1, \ldots, 1)^T \in \mathbb{R}^{10}$. Table 3 transcribes the average running best function value and its corresponding standard deviation over the runs. Although at 50 iterations the accuracy of our algorithm is worse than for the other presented algorithms, the comparison reverses at 500 iterations, where our method outperforms the state-of-the-art methods by an order of magnitude of 10. In the state-of-the-art simulated annealing methods, we observe very little improvement between 50 and 500 iterations compared to our algorithm. This reflects the well-known fact that simulated annealing methods tend to get stuck in local minimizers if the cooling scheme is not tailored to the problem, issue which our method does not seem to encounter.

# 6 Conclusions

The main focus of the paper is on a global optimization algorithm, which produces arbitrarily accurate solution with high probability. The main assumptions on the potential to be minimized were some regularity assumptions, and a log-Sobolev inequality on the Gibbs measure. This global optimization algorithm relies on an oracle sub-algorithm, which produces samples from a given Gibbs distribution.

For this oracle algorithm, we introduced a new variant of Langevin dynamics, given in System (9). These dynamics are inspired by the deterministic high-resolution differential equation presented in System (2) to tackle global optimization in nonconvex settings. Our continuous-time and discrete-time dynamics complement the classical overdamped and underdamped Langevin methods by adding another noise term. We established convergence properties of the continuous-time dynamics, showing that their invariant measure concentrates around the global minimizers of the potential $U$, and developed a practical sampling algorithm through discretization, which provably converges.

Our theoretical analysis and numerical experiments on the Rastrigin function demonstrate that the proposed method effectively navigates nonconvex landscapes and can obtain accurate solutions with high probability.

## 6.1 Open Questions

Several open questions remain for future work:

1. The parameters of the algorithm were presented in a general form, and the optimal selection of these parameters is unclear. This is also the case for the ideal combination of $(N, K)$, as suggested by Table 2.

2. The constant $C$ in Inequality (4) is not explicitly defined in Hasenpflug et al. (2024). An explicit value would enable an explicit bound in Theorem 3.1.

3. The technical Assumption B.1 might be restrictive in certain contexts. Under more general smoothness assumptions, Algorithm 1 can make use of sampling algorithms from non-smooth potentials, such as the proximal algorithm (Lee et al., 2021; Chen et al., 2022; Liang and Chen, 2023). Whence the interest in establishing a bound similar to Inequality (5), under less restrictive assumptions.

4. The cooling scheme outlined in Equation (13) is suboptimal, as evidenced by Figure 5. Further research is needed to provide a theoretical analysis of the simulated annealing variant of our algorithm. Additionally, the exploration of an adaptive scheme could be considered.

## 6.2 Acknowledgements

The authors thank the anonymous reviewers for their careful reading and insightful feedback on our manuscript.

They moreover thank Guillaume Garrigos, Huiyuan Guo, Lucas Ketels, Filip Voronine and Zepeng Wang, for our numerous meetings and their interesting and helpful insights and questions, and Mareike Hasenpflug for her help regarding her paper (Hasenpflug et al., 2024).

This research benefited from the support of the FMJH Program Gaspard Monge for optimization and operations research and their interactions with data science. We thank the Center for Information Technology of the University of Groningen for their support and for providing access to the Hábrók high performance computing cluster.

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

# A Deferred Figures

Figure 6 depicts the Rastrigin function in dimensions $d = 1$ and $d = 2$, illustrating its nonconvex nature.

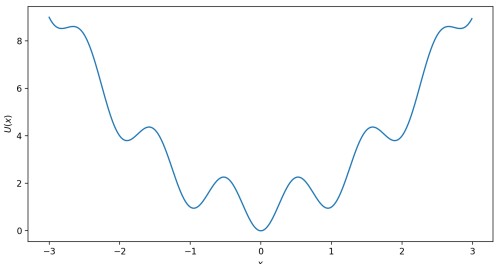 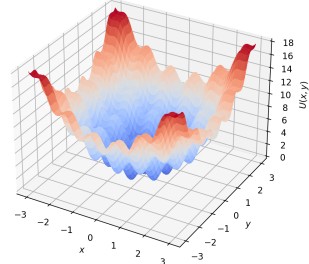

Figure 6: Rastrigin Function for $d = 1$ and $d = 2$.

# B Technical Assumptions

Equation (4) holds under the following technical assumptions:

**Assumption B.1.** We assume

1. There exists a global minimizer $\hat{x}$ of $U$ satisfying

$$\int_{\mathbb{R}^d} \|x - \hat{x}\|^2 \exp(-U(x))dx < +\infty.$$

2. $U$ is 6-times continuously differentiable on a set $F \subset \mathbb{R}^d$ such that all global minimizers are contained in the interior of $F$.

3. The Hessian of $U$ is positive definite at each global minimizer.

# C Deferred Proofs

For a probability measure $\boldsymbol{\mu} \in \mathcal{P}(\mathbb{R}^d)$, we use the notation $\boldsymbol{\mu}(f) := \mathbb{E}[f(Z)]$ where $Z \sim \boldsymbol{\mu}$ and $f$ is an integrable test function defined on $\mathbb{R}^d$. Given a process $Z_t \sim \boldsymbol{\mu}_t$, the notation $\boldsymbol{\mu}_{0|t}(f)$ denotes the function defined on $\mathbb{R}^d$ such that $\boldsymbol{\mu}_{0|t}(f)(z) := \mathbb{E}[f(Z_0)|Z_t = z]$ for any $z \in \mathbb{R}^d$.

We start by introducing a lemma that will be used throughout the proofs.

**Lemma C.1.** *Let $\boldsymbol{\mu}_t \colon \mathbb{R}_{\geq 0} \to \mathcal{P}(\mathbb{R}^d)$ be a curve. Suppose that $v_t \colon \mathbb{R}_{\geq 0} \times \mathbb{R}^d \to \mathbb{R}^d$ is a sequence of vector fields satisfying $\frac{\partial}{\partial t}\boldsymbol{\mu}_t + \nabla \cdot (\boldsymbol{\mu}_t \cdot v_t) = 0$. Then it holds that, for all $\boldsymbol{\mu}^* \in \mathcal{P}(\mathbb{R}^d)$,*

$$\frac{d}{dt}\mathrm{KL}(\boldsymbol{\mu}_t\|\boldsymbol{\mu}^*) = \mathbb{E}_{\boldsymbol{\mu}_t}\left[\left\langle \nabla \log \frac{\boldsymbol{\mu}_t}{\boldsymbol{\mu}^*}, v_t \right\rangle\right].$$

*Proof.* See (Ambrosio et al., 2005, Equation 10.1.16). □

## C.1 Proof of Theorem 4.2

1. Observe that

$$\nabla_x \boldsymbol{\mu}^{a,b} = -a\nabla U(x)\boldsymbol{\mu}^{a,b}, \quad \nabla_y \boldsymbol{\mu}^{a,b} = -by\boldsymbol{\mu}^{a,b},$$

$$\nabla_{xx}\boldsymbol{\mu}^{a,b} = -a\nabla^2 U(x)\boldsymbol{\mu}^{a,b} + a^2\|\nabla U(x)\|^2\boldsymbol{\mu}^{a,b}, \quad \nabla_{yy}\boldsymbol{\mu}^{a,b} = (-b + b^2\|y\|^2)\boldsymbol{\mu}^{a,b}.$$

As such, under the given parameters, one easily checks that

$$0 = (-a\beta + \sigma_x^2 a^2)\|\nabla U(x)\|^2 + (a - \gamma b)\nabla U(x) \cdot y + (-\alpha b + \sigma_y^2 b^2)\|y\|^2 + (\beta - \sigma_x^2 a)\operatorname{tr}(\nabla^2 U(x)) + (\alpha - \sigma_y^2 b)$$

$$= \beta \operatorname{tr}(\nabla^2)U(x) + a(-\beta\nabla U(x) + y) \cdot \nabla U(x) + \alpha + b(-\gamma\nabla U(x) - \alpha y)y$$

$$\quad + \sigma_x^2(-a\operatorname{tr}(\nabla^2 U(x)) + a^2\|\nabla U(x)\|^2) + \sigma_y^2(-b + b^2\|y\|^2)$$

$$= \frac{1}{\boldsymbol{\mu}^{a,b}}\Big(-\nabla_x \cdot ((-\beta\nabla U(x) + y)\boldsymbol{\mu}^{a,b}(x,y)) - \nabla_y \cdot ((-\gamma\nabla U(x) - \alpha y)\boldsymbol{\mu}^{a,b}(x,y)) + \sigma_x^2\operatorname{tr}(\partial_{xx}\boldsymbol{\mu}^{a,b}) + \sigma_y^2\operatorname{tr}(\partial_{yy}\boldsymbol{\mu}^{a,b})\Big).$$

It thus holds that $\boldsymbol{\mu}^{a,b}$ satisfies the Fokker-Planck Equation, and must hence be an invariant distribution.

2. The Fokker-Planck Equation associated with System (9) reads

$$\partial_t \boldsymbol{\mu}_t = -\nabla_x \cdot ((-\beta\nabla U(x) + y)\boldsymbol{\mu}_t(x,y)) - \nabla_y \cdot ((-\gamma\nabla U(x) - \alpha y)\boldsymbol{\mu}_t(x,y)) + \sigma_x^2\Delta_{xx}\boldsymbol{\mu}_t + \sigma_y^2\Delta_{yy}\boldsymbol{\mu}_t$$

$$= \nabla \cdot \left[\begin{pmatrix} \sigma_x & -1/b \\ 1/b & \sigma_y \end{pmatrix}\begin{pmatrix} a\nabla U(x) \\ by \end{pmatrix}\boldsymbol{\mu}_t\right] + \nabla \cdot \left[\begin{pmatrix} \sigma_x^2 & -1/b \\ 1/b & \sigma_y^2 \end{pmatrix}\nabla\boldsymbol{\mu}_t\right]$$

$$= \nabla \cdot \left[\begin{pmatrix} \sigma_x^2 & -1/b \\ 1/b & \sigma_y^2 \end{pmatrix}\left[-\nabla\log\boldsymbol{\mu}^{a,b} + \nabla\log\boldsymbol{\mu}_t\right]\boldsymbol{\mu}_t\right]$$

$$= \nabla \cdot \left[\begin{pmatrix} \sigma_x^2 & -1/b \\ 1/b & \sigma_y^2 \end{pmatrix}\nabla\log\left(\frac{\boldsymbol{\mu}_t}{\boldsymbol{\mu}^{a,b}}\right)\boldsymbol{\mu}_t\right].$$

As such, by Lemma C.1, we know that

$$\frac{d}{dt}\operatorname{KL}(\boldsymbol{\mu}_t\|\boldsymbol{\mu}^{a,b}) = -\mathbb{E}_{\boldsymbol{\mu}_t}\left[\left\langle\nabla\log\left(\frac{\boldsymbol{\mu}_t}{\boldsymbol{\mu}^{a,b}}\right), \begin{pmatrix} \sigma_x^2 & -1/b \\ 1/b & \sigma_y^2 \end{pmatrix}\nabla\log\left(\frac{\boldsymbol{\mu}_t}{\boldsymbol{\mu}^{a,b}}\right)\right\rangle\right]$$

$$= -\mathbb{E}_{\boldsymbol{\mu}_t}\left[\left\langle\nabla\log\left(\frac{\boldsymbol{\mu}_t}{\boldsymbol{\mu}^{a,b}}\right), \begin{pmatrix} \sigma_x^2 & 0 \\ 0 & \sigma_y^2 \end{pmatrix}\nabla\log\left(\frac{\boldsymbol{\mu}_t}{\boldsymbol{\mu}^{a,b}}\right)\right\rangle\right]$$

$$\leq -\min(\sigma_x^2, \sigma_y^2)\mathbb{E}_{\boldsymbol{\mu}_t}\left[\left\|\nabla\log\left(\frac{\boldsymbol{\mu}_t}{\boldsymbol{\mu}^{a,b}}\right)\right\|^2\right]. \tag{14}$$

In specific, if we assume a log-Sobolev inequality with coefficient $\rho$, we get

$$\frac{d}{dt}\operatorname{KL}(\boldsymbol{\mu}_t\|\boldsymbol{\mu}^{a,b}) \leq -2\rho\min(\sigma_x^2, \sigma_y^2)\operatorname{KL}(\boldsymbol{\mu}_t\|\boldsymbol{\mu}^{a,b}),$$

which, by Grönwall's Inequality, implies

$$\operatorname{KL}(\boldsymbol{\mu}_t\|\boldsymbol{\mu}^{a,b}) \leq \exp(-2\rho\min(\sigma_x^2, \sigma_y^2)t)\operatorname{KL}(\boldsymbol{\mu}_0\|\boldsymbol{\mu}^{a,b}),$$

which means we get convergence as long as $\sigma_x^2, \sigma_y^2 > 0$.

### C.2 Explicit Computations for Algorithm 2

We first note that we can rewrite System (11) in integral form as, for $t \in [kh, (k+1)h)$,

$$\begin{cases} \tilde{X}_t = \tilde{X}_{kh} - \beta(t - kh)\nabla U(\tilde{X}_{kh}) + \int_{kh}^t Y_s ds + \sqrt{2\sigma_x^2}\int_{kh}^t dB_s^x \\ \tilde{Y}_t = e^{-\alpha(t-kh)}\tilde{Y}_{kh} - \frac{\gamma}{\alpha}\left(1 - e^{-\alpha(t-kh)}\right)\nabla U(\tilde{X}_{kh}) + \sqrt{2\sigma_y^2}\int_{kh}^t e^{-\alpha(t-s)}dB_s^y. \end{cases} \tag{15}$$

Conditionally on the initial condition $(\tilde{X}_{kh}, \tilde{Y}_{kh})$, the process $(\tilde{X}_t, \tilde{Y}_t)_{kh \leq t \leq (k+1)h}$ is an Ornstein-Uhlenbeck process on $[kh, (k+1)h]$.

From now on, in this subsection, we always work implicitly conditionally to $(\tilde{X}_{kh}, \tilde{Y}_{kh})$.

In particular, $\mathcal{L}((\tilde{X}_t, \tilde{Y}_t))$ is Gaussian and it remains to compute the associated expectation and covariance matrix to fully characterize it.

We thus compute

$$\mathbb{E}[\tilde{Y}_t] = e^{-\alpha(t-kh)}\tilde{Y}_{kh} - \frac{\gamma}{\alpha}\left(1 - e^{-\alpha(t-kh)}\right)\nabla U(\tilde{X}_{kh}),$$

from which we get that

$$\mathbb{E}[\tilde{X}_t] = \tilde{X}_{kh} - \beta(t-kh)\nabla U(\tilde{X}_{kh}) + \frac{1-e^{-\alpha(t-kh)}}{\alpha}\tilde{Y}_{kh} - \frac{\gamma}{\alpha}\left(t - \frac{1-e^{-\alpha(t-kh)}}{\alpha}\right)\nabla U(\tilde{X}_{kh}).$$

Now note that the Brownian motion term for $\tilde{Y}_t$ is $\sqrt{2\sigma_y^2}\int_{kh}^t e^{-\alpha(t-s)}dB_s^y$, whereas the Brownian motion term for $\tilde{X}_t$ is

$$\sqrt{2\sigma_x^2}\int_{kh}^t dB_s^x + \sqrt{2\sigma_y^2}\int_{kh}^t\int_{kh}^r e^{-\alpha(r-s)}dB_s^y dr = \sqrt{2\sigma_x^2}\int_{kh}^t dB_s^x + \sqrt{2\sigma_y^2}\int_{kh}^t\int_s^t e^{-\alpha(r-s)}dr dB_s^y$$

$$= \sqrt{2\sigma_x^2}\int_{kh}^t dB_s^x + \sqrt{2\sigma_y^2}\int_{kh}^t\frac{1-e^{-\alpha(t-s)}}{\alpha}dB_s^y.$$

As such,

$$\mathrm{Cov}\left(\tilde{Y}_t, \tilde{Y}_t\right) = \mathbb{E}\left[\left(\tilde{Y}_t - \mathbb{E}[\tilde{Y}_t]\right)\left(\tilde{Y}_t - \mathbb{E}[\tilde{Y}_t]\right)^T\right]$$

$$= 2\sigma_y^2 \cdot \mathbb{E}\left[\left(\int_{kh}^t e^{-\alpha(t-s)}dB_s^y\right)\left(\int_{kh}^t e^{-\alpha(t-s)}dB_s^y\right)^T\right]$$

$$= 2\sigma_y^2 \cdot \left(\int_{kh}^t e^{-2\alpha(t-s)}ds\right)\cdot I_d$$

$$= \sigma_y^2 \cdot \frac{1-e^{-2\alpha(t-kh)}}{\alpha}\cdot I_d.$$

Moreover,

$$\mathrm{Cov}\left(\tilde{X}_t, \tilde{Y}_t\right) = \mathbb{E}\left[\left(\tilde{X}_t - \mathbb{E}[\tilde{X}_t]\right)\left(\tilde{Y}_t - \mathbb{E}[\tilde{Y}_t]\right)^T\right]$$

$$= \mathbb{E}\left[\left(\sqrt{2\sigma_x^2}\int_{kh}^t dB_s^x + \sqrt{2\sigma_y^2}\int_{kh}^t\frac{1-e^{-\alpha(t-s)}}{\alpha}dB_s^y\right)\left(\sqrt{2\sigma_y^2}\int_{kh}^t e^{-\alpha(t-s)}dB_s^y\right)^T\right]$$

$$= 2\sqrt{\sigma_x^2\sigma_y^2}\mathbb{E}\left[\left(\int_{kh}^t dB_s^x\right)\left(\int_{kh}^t e^{-\alpha(t-s)}dB_s^y\right)^T\right]$$

$$+ \frac{2\sigma_y^2}{\alpha}\mathbb{E}\left[\left(\int_{kh}^t 1-e^{-\alpha(t-s)}dB_s^y\right)\left(\int_{kh}^t e^{-\alpha(t-s)}dB_s^y\right)^T\right]$$

$$= \frac{2\sigma_y^2}{\alpha}\left(\int_{kh}^t(1-e^{-\alpha(t-s)})e^{-\alpha(t-s)}ds\right)\cdot I_d$$

$$= \frac{\sigma_y^2(1-e^{-\alpha(t-kh)})^2}{\alpha^2}\cdot I_d.$$

And, finally,

$$
\begin{aligned}
\text{Cov}\left(\tilde{X}_t, \tilde{X}_t\right) &= \mathbb{E}\left[\left(\tilde{X}_t - \mathbb{E}[\tilde{X}_t]\right)\left(\tilde{X}_t - \mathbb{E}[\tilde{X}_t]\right)^T\right] \\
&= \mathbb{E}\left[\left(\sqrt{2\sigma_x^2}\int_{kh}^t dB_s^x + \sqrt{2\sigma_y^2}\int_{kh}^t \frac{1-e^{-\alpha(t-s)}}{\alpha}dB_s^y\right) \cdot \right. \\
&\qquad \left. \left(\sqrt{2\sigma_x^2}\int_{kh}^t dB_s^x + \sqrt{2\sigma_y^2}\int_{kh}^t \frac{1-e^{-\alpha(t-s)}}{\alpha}dB_s^y\right)^T\right] \\
&= 2\sigma_x^2\mathbb{E}\left[\left(\int_{kh}^t dB_s^x\right)\left(\int_{kh}^t dB_s^x\right)^T\right] + 2\sigma_y^2\mathbb{E}\left[\left(\int_{kh}^t \frac{1-e^{-\alpha(t-s)}}{\alpha}dB_s^y\right)\left(\int_0^t \frac{1-e^{-\alpha(t-s)}}{\alpha}dB_s^y\right)^T\right] \\
&= 2\sigma_x^2\left(\int_{kh}^t ds\right)\cdot I_d + \frac{2\sigma_y^2}{\alpha^2}\left(\int_{kh}^t (1-e^{-\alpha(t-s)})^2 ds\right)\cdot I_d \\
&= \left(2\sigma_x^2 + \frac{\sigma_y^2}{\alpha^3}\left[2\alpha(t-kh) + 1 - e^{-2\alpha(t-kh)} - 4(1-e^{-\alpha(t-kh)})\right]\right)\cdot I_d.
\end{aligned}
$$

Selecting $t = (k+1)h$ yields the wanted result.

### C.3  Proof of Theorem 4.3

For completion, we introduce the following technical lemma:

**Lemma C.2.** *Consider the $\mathbb{R}^d$-valued random process $(Z_t)$ defined through*

$$
dZ_t = b(Z_0)dt + \sigma dW_t,
$$

*with initial condition $Z_0 \sim \boldsymbol{\nu}_0$ for some $\boldsymbol{\nu}_0 \in \mathcal{P}(\mathbb{R}^d)$. Then $\boldsymbol{\nu}_t = \mathcal{L}(Z_t)$ is a weak solution of*

$$
\partial_t \boldsymbol{\nu}_t = L_t^* \boldsymbol{\nu}_t,
$$

*where*

$$
L_t^*\boldsymbol{\eta} = -\sum_{i=1}^d \partial_i(\boldsymbol{\nu}_{0|t}(b)\boldsymbol{\eta}) + \frac{1}{2}\sum_{i,j=1}^d \partial_{i,j}\left((\sigma\sigma^T)_{i,j}\boldsymbol{\eta}\right).
$$

*Proof.* Let us consider a smooth real-valued function $f$. Then Ito's lemma together with the tower property of conditional expectation yields

$$
\mathbb{E}[f(Z_t)] = \mathbb{E}[f(Z_0)] + \int_0^t \mathbb{E}\left[\sum_{i=1}^d \mathbb{E}[b_i(Z_0)|Z_s]\partial_i f(Z_s) + \frac{1}{2}\sum_{i,j=1}^d (\sigma\sigma^T)_{i,j}\partial_{i,j}f(Z_s)\right]ds,
$$

which reads

$$
(\boldsymbol{\nu}_t - \boldsymbol{\nu}_0)(f) = \int_0^t \boldsymbol{\nu}_s\left(\sum_{i=1}^d \boldsymbol{\nu}_{0|s}(b_i)\partial_i f + \frac{1}{2}\sum_{i,j=1}^d (\sigma\sigma^T)_{i,j}\partial_{i,j}f\right)ds.
$$

Let $L$ be the differential operator such that

$$
L_t f := \sum_{i=1}^d \boldsymbol{\nu}_{0|t}(b_i)\partial_i f + \frac{1}{2}\sum_{i,j=1}^d (\sigma\sigma^T)_{i,j}\partial_{i,j}f
$$

Then

$$
(\boldsymbol{\nu}_t - \boldsymbol{\nu}_0)(f) = \int_0^t \boldsymbol{\nu}_s(L_t f)ds = \int_0^t L_t^*\boldsymbol{\nu}_s(f)ds,
$$

and differentiating yields the wanted result. $\qquad\square$

Given the parameters $\alpha, \beta, \gamma, \sigma_x^2, \sigma_y^2, a, b, \rho, L$, we define

$$
\begin{cases}
\theta & := \rho \min(\sigma_x^2, \sigma_y^2), \\
\tau & := \dfrac{a^2 L^2 (\sigma_x^4 + b^{-2})}{2 \min(\sigma_x^2, \sigma_y^2)}, \\
A & := 12 + 4\beta^2 L^2 + 4\gamma^2 L^2, \\
B & := 2\sigma_x^2 + \dfrac{12}{b} + \dfrac{4\beta^2 L}{a} + 3\sigma_y^2 + \dfrac{4\gamma^2 L}{a}, \\
\hat{B} & := 2\tau B \cdot d.
\end{cases}
\tag{16}
$$

We now introduce the more precise statement of Theorem 4.3.

**Theorem C.3.** *Assume $\boldsymbol{\mu}^{a,b}$ satisfies a log-Sobolev inequality with constant $\rho$. Assume $(\tilde{X}_t, \tilde{Y}_t)$ follows System (11), with initial distribution $(\tilde{X}_0, \tilde{Y}_0) \sim \tilde{\boldsymbol{\mu}}_0$. Moreover, assume that*

$$
h < \min\left(1, \frac{1}{\theta}, \sqrt{\frac{\theta\rho}{8\tau A}}\right).
\tag{17}
$$

*Then it holds that*

$$
\mathrm{KL}(\tilde{\boldsymbol{\mu}}_h \| \boldsymbol{\mu}^{a,b}) \leq e^{-\theta h/2} \mathrm{KL}(\tilde{\boldsymbol{\mu}}_0 \| \boldsymbol{\mu}^{a,b}) + \hat{B} h^2,
\tag{18}
$$

*and for all $K \geq 1$,*

$$
\mathrm{KL}(\tilde{\boldsymbol{\mu}}_{Kh} \| \boldsymbol{\mu}^{a,b}) \leq \exp(-\theta K h/2) \mathrm{KL}(\tilde{\boldsymbol{\mu}}_0 \| \boldsymbol{\mu}^{a,b}) + \frac{3\hat{B} h}{4\theta}.
\tag{19}
$$

*Proof.* Choose $h$ according to Equation (17), and fix some $t \leq h$, such that System (11) reduces to

$$
\begin{cases}
\tilde{X}_t = \tilde{X}_0 - \beta t \nabla U(\tilde{X}_0) + \displaystyle\int_0^t Y_s ds + \sqrt{2\sigma_x^2} \int_0^t dB_s^x \\
\tilde{Y}_t = e^{-\alpha t} \tilde{Y}_0 - \dfrac{\gamma}{\alpha}\left(1 - e^{-\alpha t}\right) \nabla U(\tilde{X}_0) + \sqrt{2\sigma_y^2} \displaystyle\int_0^t e^{-\alpha(t-s)} dB_s^y.
\end{cases}
\tag{20}
$$

Denote by $\tilde{\boldsymbol{\mu}}_t$ the joint law of $(\tilde{X}_t, \tilde{Y}_t)$. Note that in the above process, $(\tilde{X}_0, \tilde{Y}_0)$ is itself a random variable, with joint law $\tilde{\boldsymbol{\mu}}_0$. Denote by $\tilde{\boldsymbol{\mu}}_{0,t}$ the joint law of $(\tilde{X}_0, \tilde{Y}_0)$ and $(\tilde{X}_t, \tilde{Y}_t)$, and by $\tilde{\boldsymbol{\mu}}_{0|t}$ the conditional law $\mathcal{L}((\tilde{X}_0, \tilde{Y}_0)|(\tilde{X}_t, \tilde{Y}_t))$.

Applying Lemma C.2 to the process $\tilde{Z} = (\tilde{X}, \tilde{Y})$ implies that $\tilde{\boldsymbol{\mu}}_t$ satisfies

$$
\begin{aligned}
\frac{\partial}{\partial t} \tilde{\boldsymbol{\mu}}_t &= \nabla \cdot \left( \begin{pmatrix} \sigma_x^2 & -1/b \\ 1/b & \sigma_y^2 \end{pmatrix} \left[ \begin{pmatrix} a\tilde{\boldsymbol{\mu}}_{0|t}(\nabla U)(x,y) \\ by \end{pmatrix} \tilde{\boldsymbol{\mu}}_t + \nabla \tilde{\boldsymbol{\mu}}_t \right] \right) \\
&= \nabla \cdot \left( \begin{pmatrix} \sigma_x^2 & -1/b \\ 1/b & \sigma_y^2 \end{pmatrix} \left[ \begin{pmatrix} a\nabla U(x) \\ by \end{pmatrix} + \nabla \log(\tilde{\boldsymbol{\mu}}_t) + \begin{pmatrix} a\boldsymbol{\mu}_{0|t}(\nabla U)(x,y) - \nabla U(x) \\ 0 \end{pmatrix} \right] \tilde{\boldsymbol{\mu}}_t \right) \\
&= \nabla \cdot \left( \begin{pmatrix} \sigma_x^2 & -1/b \\ 1/b & \sigma_y^2 \end{pmatrix} \left[ \nabla \log \frac{\tilde{\boldsymbol{\mu}}_t}{\boldsymbol{\mu}^{a,b}} + \begin{pmatrix} a\boldsymbol{\mu}_{0|t}(\nabla U)(x,y) - \nabla U(x) \\ 0 \end{pmatrix} \right] \tilde{\boldsymbol{\mu}}_t \right).
\end{aligned}
$$

As such, the vector field given by

$$
v_t = -\begin{pmatrix} \sigma_x^2 & -1/b \\ 1/b & \sigma_y^2 \end{pmatrix} \left( \nabla \log \frac{\tilde{\boldsymbol{\mu}}_t}{\boldsymbol{\mu}^{a,b}} + \begin{pmatrix} a\tilde{\boldsymbol{\mu}}_{0|t}(\nabla U)(x,y) - \nabla U(x) \\ 0 \end{pmatrix} \right)
$$

is tangent to $(\tilde{\boldsymbol{\mu}}_t)$, and hence, by Lemma C.1, it holds that

$$
\begin{aligned}
\frac{d}{dt} \mathrm{KL}(\tilde{\boldsymbol{\mu}}_t \| \boldsymbol{\mu}^{a,b}) &= -\mathbb{E}_{\tilde{\boldsymbol{\mu}}_t} \left[ \left\langle \nabla \log \frac{\tilde{\boldsymbol{\mu}}_t}{\boldsymbol{\mu}^{a,b}}, \begin{pmatrix} \sigma_x^2 & -1/b \\ 1/b & \sigma_y^2 \end{pmatrix} \left( \nabla \log \frac{\tilde{\boldsymbol{\mu}}_t}{\boldsymbol{\mu}^{a,b}} + \begin{pmatrix} a\tilde{\boldsymbol{\mu}}_{0|t}(\nabla U)(\tilde{X}_t, \tilde{Y}_t) - \nabla U(\tilde{X}_t) \\ 0 \end{pmatrix} \right) \right\rangle \right] \\
&= -\mathbb{E}_{\tilde{\boldsymbol{\mu}}_t} \left[ \left\langle \nabla \log \frac{\tilde{\boldsymbol{\mu}}_t}{\boldsymbol{\mu}^{a,b}}, \begin{pmatrix} \sigma_x^2 & -1/b \\ 1/b & \sigma_y^2 \end{pmatrix} \nabla \log \frac{\tilde{\boldsymbol{\mu}}_t}{\boldsymbol{\mu}^{a,b}} \right\rangle \right] \\
&\quad - \mathbb{E}_{\tilde{\boldsymbol{\mu}}_t} \left[ \left\langle \nabla \log \frac{\tilde{\boldsymbol{\mu}}_t}{\boldsymbol{\mu}^{a,b}}, \begin{pmatrix} \sigma_x^2 & -1/b \\ 1/b & \sigma_y^2 \end{pmatrix} \begin{pmatrix} a\tilde{\boldsymbol{\mu}}_{0|t}(\nabla U)(\tilde{X}_t, \tilde{Y}_t) - \nabla U(\tilde{X}_t) \\ 0 \end{pmatrix} \right\rangle \right] \\
&= -\mathbb{E}_{\tilde{\boldsymbol{\mu}}_t} \left[ \left\langle \nabla \log \frac{\tilde{\boldsymbol{\mu}}_t}{\boldsymbol{\mu}^{a,b}}, \begin{pmatrix} \sigma_x^2 & 0 \\ 0 & \sigma_y^2 \end{pmatrix} \nabla \log \frac{\tilde{\boldsymbol{\mu}}_t}{\boldsymbol{\mu}^{a,b}} \right\rangle \right] \\
&\quad - \mathbb{E}_{\tilde{\boldsymbol{\mu}}_{0,t}} \left[ \left\langle \nabla \log \frac{\tilde{\boldsymbol{\mu}}_t}{\boldsymbol{\mu}^{a,b}}, \begin{pmatrix} \sigma_x^2 & -1/b \\ 1/b & \sigma_y^2 \end{pmatrix} \begin{pmatrix} a[\nabla U(\tilde{X}_0) - \nabla U(\tilde{X}_t)] \\ 0 \end{pmatrix} \right\rangle \right] \\
&\leq -\frac{\min(\sigma_x^2, \sigma_y^2)}{2} \mathrm{Fi}(\tilde{\boldsymbol{\mu}}_t \| \boldsymbol{\mu}^{a,b}) \\
&\quad + \frac{1}{2\min(\sigma_x^2, \sigma_y^2)} \mathbb{E}_{\tilde{\boldsymbol{\mu}}_{0,t}} \left[ \left\| \begin{pmatrix} \sigma_x^2 & -1/b \\ 1/b & \sigma_y^2 \end{pmatrix} \begin{pmatrix} a[\nabla U(\tilde{X}_0) - \nabla U(\tilde{X}_t)] \\ 0 \end{pmatrix} \right\|^2 \right],
\end{aligned}
$$

(21)

(22)

where the final inequality follows by Young's Inequality with coefficient $\min(\sigma_x^2, \sigma_y^2)$. Now note that

$$
\begin{aligned}
\mathbb{E}_{\tilde{\boldsymbol{\mu}}_{0,t}} \left[ \left\| \begin{pmatrix} \sigma_x^2 & -1/b \\ 1/b & \sigma_y^2 \end{pmatrix} \begin{pmatrix} a[\nabla U(\tilde{X}_0) - \nabla U(\tilde{X}_t)] \\ 0 \end{pmatrix} \right\|^2 \right] &= a^2(\sigma_x^4 + b^{-2}) \mathbb{E}_{\tilde{\boldsymbol{\mu}}_{0,t}} [\|\nabla U(\tilde{X}_0) - \nabla U(\tilde{X}_t)\|^2] \\
&\leq a^2 L^2 (\sigma_x^4 + b^{-2}) \mathbb{E}_{\tilde{\boldsymbol{\mu}}_{0,t}} [\|\tilde{X}_0 - \tilde{X}_t\|^2].
\end{aligned}
$$

As such, Equation (22) reads

$$
\frac{d}{dt} \mathrm{KL}(\tilde{\boldsymbol{\mu}}_t \| \boldsymbol{\mu}^{a,b}) \leq -\frac{\min(\sigma_x^2, \sigma_y^2)}{2} \mathrm{Fi}(\tilde{\boldsymbol{\mu}}_t \| \boldsymbol{\mu}^{a,b}) + \tau \mathbb{E}_{\tilde{\boldsymbol{\mu}}_{0,t}} [\|\tilde{X}_0 - \tilde{X}_t\|^2].
$$

(23)

Now notice that, by the integral Equations (20) and Jensen's Inequality,

$$
\begin{aligned}
\mathbb{E}_{\tilde{\boldsymbol{\mu}}_{0,t}} [\|\tilde{X}_0 - \tilde{X}_t\|^2] &= \mathbb{E}_{\tilde{\boldsymbol{\mu}}_{0,t}} \left[ \left\| \beta t \nabla U(\tilde{X}_0) + \int_0^t \tilde{Y}_s ds + \sqrt{2\sigma_x^2} \int_0^t d\tilde{B}_s \right\|^2 \right] \\
&\leq 2\beta^2 t^2 \mathbb{E}_{\tilde{\boldsymbol{\mu}}_{0,t}} [\|\nabla U(\tilde{X}_0)\|^2] + 2t \int_0^t \mathbb{E}_{\tilde{\boldsymbol{\mu}}_{0,t}} \left[ \|\tilde{Y}_s\|^2 \right] ds + 2\sigma_x^2 d \cdot t.
\end{aligned}
$$

(24)

Now observe that, for $\boldsymbol{\zeta}$ an optimal coupling between $\tilde{\boldsymbol{\mu}}_0$ and $\boldsymbol{\mu}^{a,b}$ and for $(X^{a,b}, Y^{a,b}) \sim \boldsymbol{\mu}^{a,b}$,

$$
\begin{aligned}
\mathbb{E}_{\tilde{\boldsymbol{\mu}}_0} [\|\nabla U(\tilde{X}_0)\|^2] &\leq 2\mathbb{E}_{\boldsymbol{\zeta}} [\|\nabla U(\tilde{X}_0) - \nabla U(X^{a,b})\|^2] + 2\mathbb{E}_{\boldsymbol{\mu}^{a,b}} [\|\nabla U(X^{a,b})\|^2] \\
&\leq 2L^2 \mathbb{E}_{\boldsymbol{\zeta}} [\|\tilde{X}_0 - X^{a,b}\|^2] + 2\mathbb{E}_{\boldsymbol{\mu}^{a,b}} [\|\nabla U(X^{a,b})\|^2] \\
&= 2L^2 W_0 + 2\mathbb{E}_{\boldsymbol{\mu}^{a,b}} [\|\nabla U(X^{a,b})\|^2],
\end{aligned}
$$

(25)

where $W_0 = W_2^2(\tilde{\boldsymbol{\mu}}_0, \boldsymbol{\mu}^{a,b})$. Moreover, by denoting $Z_{a,b}$ the normalization constant of $\boldsymbol{\mu}^{a,b}$,

$$
\begin{aligned}
\mathbb{E}_{\boldsymbol{\mu}^{a,b}}[\|\nabla U(x)\|^2] &= \iint_{\mathbb{R}^d \times \mathbb{R}^d} \nabla U(x) \cdot \nabla U(x) d\boldsymbol{\mu}^{a,b}(x,y) \\
&= \frac{1}{Z_{a,b}} \int_{\mathbb{R}^d} e^{-b\|y\|^2/2} \int_{\mathbb{R}^d} \nabla U(x) \cdot \nabla U(x) e^{-aU(x)} dx dy \\
&= \frac{-1}{Z_{a,b}a} \int_{\mathbb{R}^d} e^{-b\|y\|^2/2} \int_{\mathbb{R}^d} \nabla U(x) \cdot \nabla e^{-aU(x)} dx dy \\
&= \frac{1}{Z_{a,b}a} \int_{\mathbb{R}^d} e^{-b\|y\|^2/2} \int_{\mathbb{R}^d} \Delta U(x) e^{-aU(x)} dx dy \\
&= \frac{1}{a} \mathbb{E}_{\boldsymbol{\mu}^{a,b}}[\Delta U(X^{a,b})] \\
&\leq \frac{dL}{a},
\end{aligned}
\tag{26}
$$

where the third-to-last equality follows from integration by parts, and the final inequality follows from $L$-smoothness. Plugging this into (25) then yields

$$
\mathbb{E}_{\tilde{\boldsymbol{\mu}}_0}[\|\nabla U(\tilde{X}_0)\|^2] \leq 2L^2 W_0 + \frac{2dL}{a},
\tag{27}
$$

which, replacing in Equation (24), gives

$$
\mathbb{E}_{\tilde{\boldsymbol{\mu}}_{0,t}}\left[\|\tilde{X}_0 - \tilde{X}_t\|^2\right] \leq 2\beta^2 t^2 \left(2L^2 W_0 + \frac{2dL}{a}\right) + 2t \int_0^t \mathbb{E}_{\tilde{\boldsymbol{\mu}}_{0,t}}\left[\|\tilde{Y}_s\|^2\right] ds + 2\sigma_x^2 d \cdot t.
\tag{28}
$$

By Equations (20)

$$
\begin{aligned}
\mathbb{E}_{\tilde{\boldsymbol{\mu}}^{(0,s)}}\left[\|\tilde{Y}_t\|^2\right] &\leq 3\mathbb{E}_{\tilde{\boldsymbol{\mu}}_0}\left[\|\tilde{Y}_0\|^2\right] + 3\gamma^2 \mathbb{E}_{\tilde{\boldsymbol{\mu}}_0}\left[\left\|\int_0^s \nabla U(\tilde{X}_0) dr\right\|^2\right] + 6\sigma_y^2 \mathbb{E}_{\tilde{\boldsymbol{\mu}}^{(0,s)}}\left[\left\|\int_0^s e^{-\alpha(s-r)} dB_r^y\right\|^2\right] \\
&\leq 6W_0 + 6\mathbb{E}_{\boldsymbol{\mu}^{a,b}}\left[\|Y^{a,b}\|^2\right] + 3\gamma^2 s^2 \mathbb{E}_{\tilde{\boldsymbol{\mu}}_0}\left[\|\nabla U(\tilde{X}_0)\|^2\right] + 6\sigma_y^2 d \cdot s \\
&\leq 6W_0 + 6\mathbb{E}_{\boldsymbol{\mu}^{a,b}}\left[\|Y^{a,b}\|^2\right] + 3\gamma^2 s^2 \left(2L^2 W_0 + \frac{2dL}{a}\right) + 6\sigma_y^2 d \cdot s.
\end{aligned}
\tag{29}
$$

Now we realize that analogous computations to (26) give that

$$
\mathbb{E}_{\boldsymbol{\mu}^{a,b}}\left[\|Y^{a,b}\|^2\right] \leq \frac{d}{b},
$$

which, plugged into (29), yields, after rearranging the terms,

$$
\mathbb{E}_{\tilde{\boldsymbol{\mu}}_t}\left[\|\tilde{Y}_t\|^2\right] \leq 6W_0 \left(1 + \gamma^2 L^2 t^2\right) + 6\frac{d}{b} + 6d \cdot t \left(\frac{\gamma^2 Lt}{a} + \sigma_y^2\right).
$$

Returning to Equation (28), we obtain

$$
\begin{aligned}
\mathbb{E}_{\tilde{\boldsymbol{\mu}}_{0,t}}\left[\|\tilde{X}_0 - \tilde{X}_t\|^2\right] &\leq 2\beta^2 t^2 \left(2L^2 W_0 + \frac{2dL}{a}\right) + 2\sigma_x^2 d \cdot t \\
&\quad + 2t \int_0^t \left(6W_0 \left(1 + \gamma^2 L^2 s^2\right) + 6\frac{d}{b} + 6ds \left(\frac{2\gamma^2 Ls}{a} + \sigma_y^2\right)\right) ds \\
&= 2\beta^2 t^2 \left(2L^2 W_0 + \frac{2dL}{a}\right) + 2\sigma_x^2 d \cdot t \\
&\quad + 2t \left(6W_0 \left(t + \frac{\gamma^2 L^2 t^3}{3}\right) + \frac{6d \cdot t}{b} + 2d \cdot t^3 \frac{2\gamma^2 L}{a} + 3d \cdot t^2 \sigma_y^2\right) \\
&= W_0 \cdot \left[12t^2 + 4\beta^2 L^2 t^2 + 4\gamma^2 L^2 t^3\right] + d \cdot \left[2\sigma_x^2 t + \frac{12t^2}{b} + \frac{4\beta^2 L}{a} \cdot t^2 + 3\sigma_y^2 \cdot t^3 + \frac{4\gamma^2 L}{a} \cdot t^4\right].
\end{aligned}
$$

Using that $t \leq h < 1$ and recalling the notation defined in Equation (16), we obtain

$$\mathbb{E}_{\tilde{\boldsymbol{\mu}}_{0,t}} \left[ \left\| \tilde{X}_0 - \tilde{X}_t \right\|^2 \right] \leq W_0 \cdot A \cdot t^2 + d \cdot B \cdot t.$$

Combining the above with (23), we obtain

$$\frac{d}{dt} \operatorname{KL}(\tilde{\boldsymbol{\mu}}_t \| \boldsymbol{\mu}^{a,b}) \leq -\frac{\min(\sigma_x^2, \sigma_y^2)}{2} \operatorname{Fi}(\tilde{\boldsymbol{\mu}}_t \| \boldsymbol{\mu}^{a,b}) + \tau \cdot W_0 \cdot A \cdot t^2 + \tau \cdot d \cdot B \cdot t.$$

Applying a log-Sobolev inequality, we obtain

$$\frac{d}{dt} \operatorname{KL}(\tilde{\boldsymbol{\mu}}_t \| \boldsymbol{\mu}^{a,b}) \leq -\theta \operatorname{KL}(\tilde{\boldsymbol{\mu}}_t \| \boldsymbol{\mu}^{a,b}) + \tau A \cdot W_0 \cdot t^2 + \tau B \cdot d \cdot t.$$

Rearranging the terms yields

$$\frac{d}{dt} e^{\theta t} \operatorname{KL}(\tilde{\boldsymbol{\mu}}_t \| \boldsymbol{\mu}^{a,b}) \leq e^{\theta t} \tau A \cdot W_0 \cdot t^2 + e^{\theta t} \tau B \cdot d \cdot t.$$

As $t \leq h$,

$$\frac{d}{dt} e^{\theta t} \operatorname{KL}(\tilde{\boldsymbol{\mu}}_t \| \boldsymbol{\mu}^{a,b}) \leq e^{\theta t} \tau A \cdot W_0 \cdot h^2 + e^{\theta t} \tau B \cdot d \cdot h,$$

which we integrate from $0$ to $h$ in $t$, yielding

$$\begin{aligned}
e^{\theta h} \operatorname{KL}(\tilde{\boldsymbol{\mu}}^{(h)} \| \boldsymbol{\mu}^{a,b}) - \operatorname{KL}(\tilde{\boldsymbol{\mu}}_0 \| \boldsymbol{\mu}^{a,b}) &\leq \frac{e^{\theta h} - 1}{\theta} \tau A \cdot W_0 \cdot h^2 + \frac{e^{\theta h} - 1}{\theta} \tau B \cdot d \cdot h \\
&\leq 2\tau A \cdot W_0 \cdot h^3 + 2\tau B \cdot d \cdot h^2,
\end{aligned}$$

where we used $e^c \leq 1 + 2c$ for $0 < c < 1$ for $c = \theta h$. Using Talagrand's Inequality (Talagrand, 1996) and denoting $\operatorname{KL}_0 = \operatorname{KL}(\tilde{\boldsymbol{\mu}}_0 \| \boldsymbol{\mu}^{a,b})$ gives us that

$$e^{\theta h} \operatorname{KL}(\tilde{\boldsymbol{\mu}}^{(h)} \| \boldsymbol{\mu}^{a,b}) - \operatorname{KL}_0 \leq \frac{4}{\rho} \tau A \cdot \operatorname{KL}_0 \cdot h^3 + 2\tau B \cdot d \cdot h^2,$$

This implies that

$$\operatorname{KL}(\tilde{\boldsymbol{\mu}}^{(h)} \| \boldsymbol{\mu}^{a,b}) \leq e^{-\theta h} \operatorname{KL}_0 \cdot \left( 1 + \frac{4}{\rho} \tau A \cdot h^3 \right) + 2 e^{-\theta h} \tau B \cdot d \cdot h^2,$$

which implies (18), using (17) and the fact that $\ln(x) \leq x - 1$ for $x > 0$.

Iteratively applying the result of (18), we obtain, where $\operatorname{KL}_k := \operatorname{KL}(\tilde{\boldsymbol{\mu}}_{kh} \| \boldsymbol{\mu}^{a,b})$ for all $k$,

$$\begin{aligned}
\operatorname{KL}_K &\leq e^{-\theta h/2} \operatorname{KL}_{K-1} + \hat{B} h^2 \\
&\leq e^{-2\theta h/2} \operatorname{KL}_{K-2} + (e^{-\theta h/2} + 1) \hat{B} h^2 \\
&\leq \cdots \\
&\leq e^{-K\theta h/2} \operatorname{KL}_0 + \left( \sum_{k=0}^{K-1} e^{-k\theta h/2} \right) \hat{B} h^2,
\end{aligned}$$

which, by bounding the finite sum by an infinite sum and using $e^{-c} \leq 1 - \frac{2}{3} c$ for $0 < c < 1$, yields (19). $\quad\square$

