# OpenReview forum: "Global Optimization Algorithm through High-Resolution Sampling"
_TMLR — Accepted by TMLR_

### Review · Reviewer_WeR4 · 2025-07-05

**Summary Of Contributions:**

The paper makes two key theoretical advances in nonconvex global optimization under a log-Sobolev inequality assumption: first, it introduces a modular “global optimization” meta‐algorithm that, by drawing $N$ i.i.d. samples from a Gibbs distribution $\mu_a$, provably finds an $\varepsilon$-accurate global minimizer of a smooth nonconvex $U$ with high probability.

Second, it proposes a novel “high-resolution” Langevin sampler both in continuous form (a noisy first-order system inspired by the accelerated deterministic ODE $ \ddot x+\alpha\dot x+\beta\nabla^2U\,\dot x+\gamma\nabla U=0 $) and an exact Gaussian discretizatio that converges exponentially fast in KL divergence to the Gibbs target whenever $μ_{a,b}$ satisfies an LSI (Thm 4.2).

Their experiments on the Rastrigin benchmark, tested at dimensions 10 and 20, exhibit empirical convergence rates that align closely with the theoretical guarantees.

**Audience:**

Yes

**Broader Impact Concerns:**

The work is largely theoretical and methodological. No immediate ethical red-flags arise.

**Claims And Evidence:**

Yes

**Requested Changes:**

Clarification needed: You assume a single constant $\rho>0$ works for every Gibbs measure
 Does it make sense that $\rho$ (log-Sobolev constant) is independent of both $a$ and  $b$ (Gibbs measure parameters)?

If possible (not blocking): An ablation study isolating the effect of (i) inertia, (ii) Hessian-driven noise, and (iii) discretization bias so we really know which component is doing the “heavy lifting.”

**Strengths And Weaknesses:**

Strength:
- Generally well-written, logical structure, extensive related-work coverage.
- Connects high-resolution ODE analysis (classically convex deterministic) with non-convex sampling, opening a new avenue for accelerated MCMC.
- The finite-time probabilistic global-optimality guarantee is a novel approach as far as I am aware.

Weekness:
- Acceleration remains unproved: rates in Cor. 4.5 match, but do not beat, those of ULA/OLA; the main motivation of the paper leads to a plausibility argument.

---

### Review · Reviewer_4QFu · 2025-07-17

**Summary Of Contributions:**

The authors consider global optimization of a nonconvex function U, under several assumptions on U, including a logarithmic Sobolev inequality on the Gibbs measure mu^ab \propto exp(-aU-b||y||^2/2) for all a>a0. Assuming access to an approximation tilde{mu} which has KL divergence with mu^a less than epsilon^2/18, the simple "optimization algorithm" of taking the best of N samples from tilde{mu} gets close to the global optimum with high probability 1-delta. For actually sampling from tilde{mu} they propose "high resolution Langevin dynamics" which can be viewed as a SDE version of ODE-type continuous representations of optimization algorithms. They prove exponential rate convergence of both the continuous and discretized versions. Finally promising numerical results are presented on the Rastrigin function.

**Audience:**

Yes

**Claims And Evidence:**

Yes

**Requested Changes:**

The numerical results are reasonable. Figures 1 and 3 should presumably be one figure. Although it doesn't connect as cleanly to the theory, it would be helpful to see something like the median error over runs as well as the probability of hitting a specific accuracy epsilon (e.g. as a supplemental fig).

Tables 1 and 2 have too many significant figures, and it is not immediately obvious that these are values of the objective function.

**Strengths And Weaknesses:**

This paper is a bit theoretically heavy for me but I follow the general argument and believe there are interesting results here. It would be helpful if some intuition as to what type of functions follow a log-Sobolev inequality, since the formal definition is obtuse (that the KL is bounded by a constant times the relative Fisher information). It would also help to give parameters informative names: e.g. a is an inverse temperature, epilson is an error tolerance (optimality gap?).

---

### Review · Reviewer_VURz · 2025-08-01

**Summary Of Contributions:**

The paper proposes a method for global optimization of non-convex functions. It samples from the Gibbs measure associated with the function using a “high resolution Langevin dynamics”, which is new to this work. Theoretical guarantees are given when the Gibbs measure satisfies a log-Sobolev inequality, and it is assessed on an empirical benchmark.

**Audience:**

Yes

**Broader Impact Concerns:**

None.

**Claims And Evidence:**

Yes

**Requested Changes:**

- Hopefully, my concerns have been clearly stated in the “Strengths and Weaknesses” section.
- I think a clearer understanding of why the high-resolution algorithm is chosen would be important. Additionally, the authors should attempt harder to show theoretical improvements over other sampling algorithms.
- The experimental results are narrow and hence unconvincing. I hope the authors can add more.

** Minor typos **
Page 3: exhibt -> exhibit
Page 11: in specific -> specifically

**Strengths And Weaknesses:**

***Strengths***

- The method makes sense and the writing is quite clear.
- The method seems to perform decently well on the example.

***Weaknesses***

- While interesting, I believe that the strategy of sampling from a Gibbs measure to optimize non-convex functions has been well-known, and relates to annealing and other similar algorithms, as discussed in comparisons and related work. It is unclear to me why this particular architecture might do better.
- It is not clear why just running the standard Langevin Monte Carlo to the Gibbs measure (with some well-chosen temperature) would not work here. Why do we want the high-resolution algorithm? It does not seem to clearly be better than the overdamped Langevin in practice. Furthermore, there are many possible tuning parameters, which makes the comparison unclear. Finally, what about Hamiltonian Monte Carlo methods?
- While the high-resolution algorithm is interesting, the theoretical analysis strongly resembles that for the overdamped/underdamped Langevin and does not show any improvement in theory.
- In the theoretical analysis, it seems important that the Gibbs measure has a good log-Sobolev constant. However, this is unlikely to occur when the function is non-convex. As a result, I do not find this theoretical result very compelling.
- There should be more experimental results.

Overall, I do not find the theoretical results to be too surprising or impactful. Nonetheless, the paper is well-written and rigorous.

---

> ### Author Response · Authors · 2025-08-14
> **Response to review**
>
> Dear reviewer,
>
> We appreciate your careful reading of our manuscript, and the constructive feedback you have provided. In what follows, we explain how we plan on addressing your concerns, and the main modifications this would entail.
>
> **On our proposed algorithms.** Our work presents two main contributions:
> 1) **A global optimization framework (Algorithm 1):** We formalize a commonly referred to but unproven connection between sampling from a Gibbs measure and global optimization. This is a key theoretical contribution and is addressed by our Theorem 3.1, which provides the foundation for our approach. The global optimization algorithm is based on a generic oracle sampler, which allows for flexibility as the oracle can be any method with a sampling guarantee, such as Langevin Monte Carlo or Hamiltonian Monte Carlo. For instance, Figure 2 implements the previously known overdamped and underdamped Langevin algorithms, as well as our high-resolution Langevin algorithm.
> 2) **The high-resolution Langevin algorithm (Algorithm 2):** Another original feature of our work is that it highlights and exploits the link between continuous-time deterministic dynamics (which is currently the subject of active research in the field of convex optimization) and stochastic dynamics (mainly considered for the simulation of Gibbs measures). This link allows us to propose an original stochastic dynamics (the so-called ''High-resolution Langevin dynamics'' Equation (9)). This algorithm is inspired by the deterministic high-resolution ODE, which is known to accelerate in the convex setting, see below. We prove the convergence of the proposed ''High-resolution Langevin dynamics'' both in continuous and discrete time, (see Theorems 4.2 and 4.3). While we do not observe acceleration over the overdamped Langevin dynamics in our toy example, nor theoretically nor empirically, this is comparable to initial findings on the deterministic ODE, and we do not consider this a failure. We will make this point clearer in the revised paper to ensure the motivation behind the algorithm is well-understood. Ultimately, we hope this work will spark further research into developing accelerated methods for nonconvex optimization, building on the success seen in the convex setting.
>
> **On the log-Sobolev constant.** We appreciate your point regarding the potential for a small log-Sobolev constant, and this concern was also raised by other reviewers. To address this, we will enhance Remark 2.3 such that it includes quantitative bounds on the log-Sobolev constant. This will provide readers with a better understanding of how the constant behaves in worst-case scenarios, giving them a verifiable method for checking Assumption 2.2 and providing a practical sense of its potential magnitude.
>
> **On experimental results.** We will incorporate additional information requested by other reviewers into the existing experiments. We believe these additions will provide the necessary insights without requiring a complete overhaul of the experimental section.
>
> **On the high-resolution ODE.** The Hessian-driven damping term in the high-resolution ODE was introduced in [Alvarez et al, JMPA, 2002], and then [Attouch et al, JDE 2016], to mitigate the oscillations in second order dynamics. The algorithmic consequences of this, including the connection with Nesterov's method, were investigated independently in  [Attouch et al, MAPR 2022] and [Shi et al, MAPR 2022]. The theoretical convergence rates obtained for the resulting algorithmic framework were comparable to those established for Nesterov's method in the convex case. However, in the strongly convex case, the theoretical guarantees were (orders of magnitude) more conservative. Nevertheless, this motivated intense research within the convex optimization community, and the high-resolution ODE seems to be displacing the classical ODE studied in [Su et al, JMLR 2026] and [Attouch et al, MAPR 2018] (which corresponds to the overdamped Langevin system) as the preferred continuous-time model for Nesterov's acceleration. Among other reasons, (1) the high-resolution model extends better to the nonsmooth setting, and (2) it captures the linear convergence rates of FISTA and the Optimized Gradient Method in the strongly convex (or P\L) case, **while the overdamped one does not**. Although our method's performance is comparable to those for the over/underdamped Langevin systems, the experience in the convex setting calls for further investigation. We believe our work is an important first step in this direction. We will elaborate further in the revised version.
>
> Finally, we would like to reiterate our gratitude for the time spent on our manuscript and the valuable feedback provided. Please do not hesitate to reach out again for more details.
>
> Faithfully Yours,
>
> The authors

---

### Author Response · Authors · 2025-09-10
**Revision of Manuscript**

Dear Action Editor, Dear Reviewers,

We would like to thank the reviewers for the reading of our manuscript and their insightful feedback on it. We have uploaded a revised version of the manuscript, incorporating the comments received.

Specifically, in the revised version, we have included more details on the relevance of the high-resolution differential equation, and more details on the log-Sobolev assumption and quantitative results on the associated constant. Moreover, we have updated the numerical section, to include some more data and updated the description and captions around the tables and figures. We believe these updates address all points raised by the reviewers.

We are grateful for the time and effort put into our submission, which has significantly strengthened this work. We hope the manuscript is now acceptable for publication at TMLR.

Faithfully Yours,

The authors

---

### Decision · Action_Editor_RTDQ · 2025-09-09

**Recommendation:** Accept with minor revision

**Additional Comments:**

All reviewers voted that the paper supports its claims with evidence and the paper would be of interest to some members of the TMLR audience. There are some valid concerns raised by Reviewer VURz, which the authors agreed to address. I'm suggesting a minor revision to take into account the feedback from the reviewers and fix any remaining issues. The authors are also invited to include more experiments to make the paper more interesting to the practitioners.

**Audience:**

Yes

**Audience Explanation:**

The paper is written on two fairly popular topics in the machine learning and optimization literature: nonconvex minimization and sampling. The results would be of potential interest to anyone working in these fields.

**Claims And Evidence:**

Yes

**Claims Explanation:**

The paper, which is explicitly a theory-focused work, presents a result on a global convergence for minimizing a nonconvex smooth potential. It states the results clearly and presents them together with the assumptions and limitations. The theory is supported by numerical experiments in a toy setting. While the results are not groundbreaking, they are rigorous and can be understood by those interested in the topic.